# Metabolic tagging of extracellular vesicles and development of enhanced extracellular vesicle based cancer vaccines

Rimsha Bhatta[1], Joonsu Han [1], Yusheng Liu[1], Yang Bo[1], David Lee[1], Jiadiao Zhou[1], Yueji Wang[1,2], Erik Russell Nelson [3,4,5,6], Qian Chen [1,4,7], Xiaojia Shelly Zhang [2,8,9], Wael Hassaneen[10,11] & Hua Wang [1,3,4,5,7,10,12] ✉

As key mediators of cellular communication, extracellular vesicles (EVs) have been actively explored for diagnostic and therapeutic applications. However, effective methods to functionalize EVs and modulate the interaction between EVs and recipient cells are still lacking. Here we report a facile and universal metabolic tagging technology that can install unique chemical tags (e.g., azido groups) onto EVs. The surface chemical tags enable conjugation of molecules via efficient click chemistry, for the tracking and targeted modulation of EVs. In the context of tumor EV vaccines, we show that the conjugation of toll-like receptor 9 agonists onto EVs enables timely activation of dendritic cells and generation of superior antitumor CD8+ T cell response. These lead to 80% tumor-free survival against E.G7 lymphoma and 33% tumor-free survival against B16F10 melanoma. Our study yields a universal technology to generate chemically tagged EVs from parent cells, modulate EV-cell interactions, and develop potent EV vaccines.

Extracellular vesicles (EVs) inherit various cellular contents[1–3] and play a critical role in intercellular communication. Extensive effort has been made to decipher the role of EVs in different diseases as well as exploring their diagnostic and therapeutic applications[1–3]. These often necessitate the functionalization of EVs for tracking and targeting purposes. As the density of proteins expressed on the cell membrane is often in the range of $10^3–10^6$ per cell[4–7], genetic expression methods can potentially introduce up to $10^6$ proteins to the parent cell, but only a small quantity are inherited by the cell-secreted EVs (10,000-fold surface area difference between a cell and an EV), let alone the complexity and varied efficiency of genetic transfection[8–10]. In principle, EVs may also bear amine-bearing proteins that are inherited from the parent cell for surface functionalization using amine-carboxyl chemistry[11–14], but the number of surface amine groups per EV could be minimal. Here we show that metabolic glycan labeling of parent cells, which can introduce $10^8–10^9$ chemical tags (e.g., azido groups) in the form of glycoproteins and glycolipids to the membrane of each cell[15–19], can generate chemically tagged EVs (Fig. 1a). We demonstrate that this EV tagging approach is universally applicable to EVs secreted by various types of cancer cells, mesenchymal stem cells (MSCs), dendritic cells (DCs), and T cells (Figs. 2–3). The surface chemical tags enable in vitro and in vivo tracking and targeting of EVs, and surface conjugation and display of molecules of interest via efficient click chemistry (Fig. 1a)[20–22].

[1]Department of Materials Science and Engineering, University of Illinois at Urbana-Champaign, Urbana, IL 61801, USA. [2]Department of Mechanical Science and Engineering, University of Illinois Urbana-Champaign, Urbana, IL 61801, USA. [3]Cancer Center at Illinois (CCIL), Urbana, IL 61801, USA. [4]Beckman Institute for Advanced Science and Technology, University of Illinois at Urbana-Champaign, Urbana, IL 61801, USA. [5]Institute for Genomic Biology, University of Illinois at Urbana-Champaign, Urbana, IL 61801, USA. [6]Department of Molecular and Integrative Physiology, University of Illinois at Urbana-, Champaign, IL, USA. [7]Materials Research Laboratory, University of Illinois at Urbana-Champaign, Urbana, IL 61801, USA. [8]Department of Civil and Environmental Engineering, University of Illinois at Urbana-Champaign, Urbana, IL 61801, USA. [9]National Center for Supercomputing Applications, Urbana, IL 61801, USA. [10]Carle College of Medicine, University of Illinois at Urbana-Champaign, Urbana, IL 61801, USA. [11]Carle Foundation Hospital, Urbana, IL 61801, USA. [12]Department of Bioengineering, University of Illinois at Urbana-Champaign, Urbana, IL 61801, USA. ✉e-mail: huawang3@illinois.edu

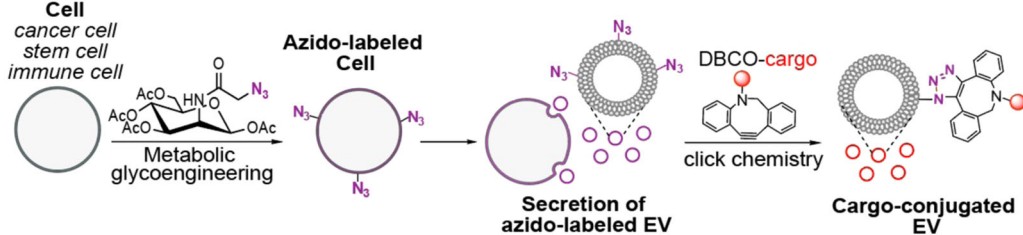

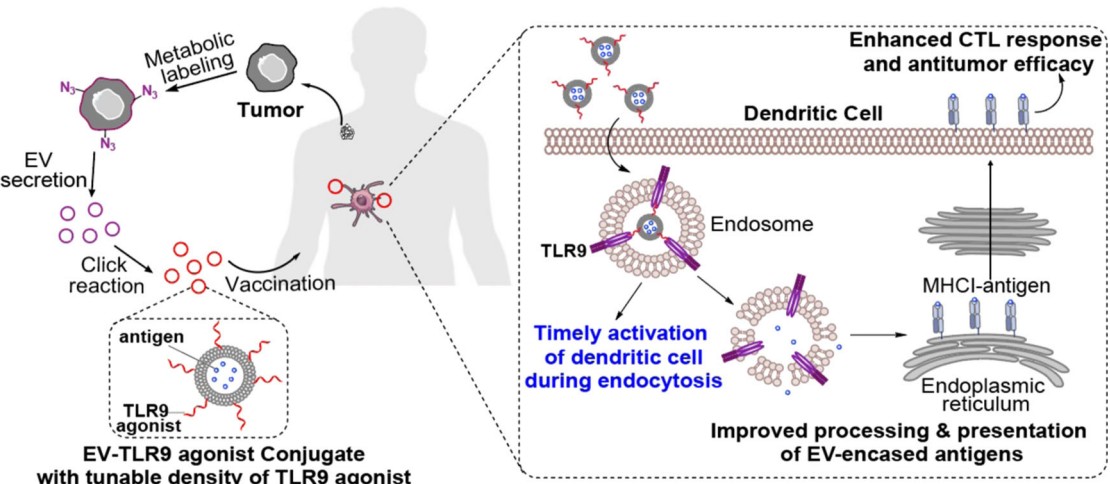

**Fig. 1 | Metabolic tagging and targeting of EVs and its use for the development of next-generation EV vaccines. a** Cells can be metabolically labeled with chemical tags (e.g., azido groups) via metabolic glycoengineering processes of unnatural sugars, for subsequent secretion of azido-tagged EVs. The azido-labeled EVs can then mediate conjugation of DBCO-cargo via efficient and bioorthogonal click chemistry, for in vitro and in vivo tracking and targeting of EVs. **b** Development of next-generation EV vaccines by orchestrating the interaction between EVs and

dendritic cells (DCs). Cells (e.g., tumor cells) from patients can be metabolically labeled to secrete azido-tagged EVs, for subsequent conjugation of TLR9 agonists via click chemistry. Upon in vivo administration, TLR9 agonist-conjugated EVs can be internalized by DCs via endosomes where TLR9 is present. The binding of TLR9 agonist on the surface of EVs to TLR9 on endosomes can stimulate DCs in a timely manner, leading to improved processing and presentation of EV-encased antigens. As a result, enhanced CTL response and antitumor efficacy can be achieved.

One pivotal role of EVs is to transport molecular signals between cells, for which the processing of EVs by the recipient cell is a critical step. For example, tumor EVs can be endocytosed by antigen presenting cells (e.g., DCs), followed by the processing and presentation of the encased antigens by DCs for subsequent priming of antigen-specific T cells[23–26]. Regarded as a safer source of tumor antigens than conventionally used dead tumor cells or tumor lysates, tumor EVs have demonstrated the ability to induce antitumor cytotoxic T lymphocyte (CTL) response in clinical trials[27,28]. However, the resultant CTL response and therapeutic efficacy are still modest[29–32], likely due to the low abundance of tumor antigens in EVs and sub-optimal activation of DCs for the processing and presentation of EV-encased antigens[31–33]. Indeed, tumor EVs alone might impair the maturation of DCs[33,34]. The incorporation of adjuvants (e.g., alum) that can activate DCs has become standard practice for improving conventional vaccines. However, simple mixing with adjuvants has failed to significantly enhance the antitumor efficacy of tumor EV vaccines[30,35]. These issues motivate the development of new approaches that can well integrate tumor EVs and adjuvants for optimal modulation of DCs and elicitation of CTL response.

We envision that the ability to quantitatively conjugate and display a high number of molecules on the surface of EVs, via our metabolic tagging and targeting approach, will enable the orchestration of the interaction between EVs and the recipient cells. As EVs are endocytosed by DCs via endosomes where TLR3, TLR7, TLR8, and TLR9 are present[23–26,36,37], we hypothesize that the display of sufficient TLR3/7/8/9

agonists on EVs can timely stimulate TLRs on the surface of intracellular endosomes, resulting in the improved activation of DCs during the endocytosis of tumor EVs (Fig. 1b). As a result, the processing and presentation of EV-encased antigens and the overall CTL response and antitumor efficacy can be enhanced. This ability to fine tune the interaction between EVs and antigen presenting cells holds promise to improve the therapeutic efficacy of tumor EV vaccines.

## Results

### Generation of chemically tagged EVs via metabolic glycan labeling

To demonstrate whether metabolically labeled cells can secrete azido-labeled EVs, we first synthesized tetraacetyl *N*-azidoacetylmannosamine (Ac₄ManAz), a common metabolic labeling agent, and used it for metabolic labeling of various types of cells. 4T1 breast cancer cells, LS174T colon cancer cells, GL261 glioblastoma cells, or BxPC-3 pancreatic cancer cells were treated with Ac₄ManAz for three days and further incubated with DBCO-Cy5 for 30 min. Uniform and bright Cy5 fluorescence signal was observed on the surface of 4T1, LS174T, GL261, and BxPC-3 cells treated with Ac₄ManAz, while control cells without Ac₄ManAz treatment showed minimal Cy5 signal (Fig. 2a–d). Flow cytometry analysis confirmed the much higher Cy5 fluorescence intensity of Ac₄ManAz-treated cells than untreated cells (Fig. S1a–d), demonstrating the successful metabolic labeling of cells with azido groups. EVs secreted by Ac₄ManAz-treated 4T1 cells or control

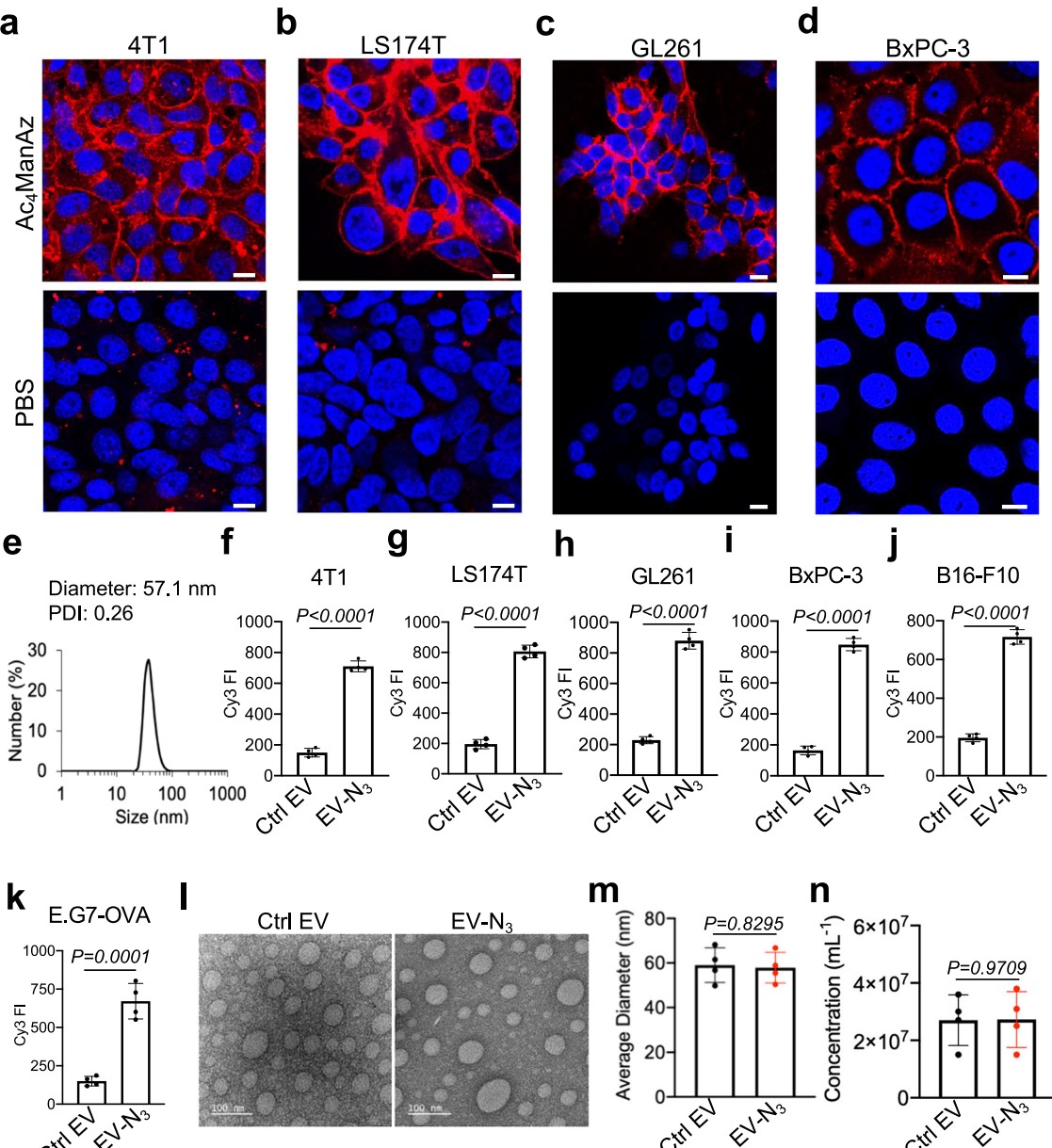

**Fig. 2 | Metabolic glycan labeling of cancer cells generates chemically tagged EVs.** CLSM images of (**a**) 4T1 breast cancer cells, (**b**) LS174T colon cancer cells, (**c**) GL261 glioblastoma cells, and (**d**) BxPC-3 pancreatic cancer cells, respectively, after treated with PBS or $Ac_4ManAz$ for three days and incubated with DBCO-Cy5 (red) for 30 min. Cell nuclei were stained with DAPI (blue). Scale bar: 10 μm. **e** Size distribution of 4T1-derived EVs. **f–k** Cell-derived EVs were isolated and stained with DBCO-Cy3, prior to the measurement of Cy3 fluorescence intensity (FI) ($n = 4$). Shown are mean Cy3 FI of EVs from $Ac_4ManAz$-treated or untreated (**f**) 4T1 cells, (**g**) LS174T cells, (**h**) GL261 cells, (**i**) BxPC-3 cells, (**j**) B16-F10 cells, and (**k**) E.G7-OVA cells. **l** TEM imaging of EVs secreted by untreated or $Ac_4ManAz$-treated E.G7-OVA cells. **m** Average diameter ($n = 4$) and (**n**) concentration of EVs secreted by $Ac_4ManAz$-treated or untreated E.G7-OVA cells ($n = 4$). All the numerical data are presented as mean ± SD (two-tailed Welch's $t$ test was used; $0.01 < *P \leq 0.05$; $**P \leq 0.01$; $***P \leq 0.001$; $****P \leq 0.0001$). Source data are provided as a Source Data file.

4T1 cells were then collected from the culture medium via a combination of ultracentrifugation and size exclusion chromatography (Fig. 2e), and incubated with DBCO-Cy3 for 30 min for the detection of surface azido groups. Compared to EVs isolated from untreated 4T1 cells, EVs from $Ac_4ManAz$-treated 4T1 cells showed significantly higher Cy3 fluorescence intensity (Fig. 2f), indicating the presence of azido groups on the surface of EVs. Similarly, EVs isolated from $Ac_4ManAz$-treated LS174T, GL261, BxPC-3, B16F10, and E.G7-OVA cells also showed higher Cy3 fluorescence intensity than EVs from untreated cells (Fig. 2g–k). ~4400 DBCO-fluorophore can be conjugated to each azido-tagged E.G7-OVA EV, indicating the presence of >4400 azido groups per EVs (Fig. S3a). These experiments demonstrated that $Ac_4ManAz$ can metabolically label cancer cells with azido

groups and the azido-labeled cells can gradually secrete azido-tagged EVs. It is noteworthy that EVs from $Ac_4ManAz$ or PBS-treated cells showed negligible changes in morphology, average diameter, and number, as determined by transmission electron microscopy, dynamic light scattering, and nanoparticle tracking system, respectively (Fig. 2l–n, Fig. S4a-d), ruling out the impact of metabolic glycan labeling on the EV secretion process.

**Metabolic tagging of MSC and immune cell-derived EVs**
We next studied whether the EVs tagging approach can be universally applied to other types of cells including MSCs, DCs, and T cells. MSCs treated with $Ac_4ManAz$ for three days and then incubated with DBCO-Cy5 showed significantly enhanced Cy5 fluorescence intensity than

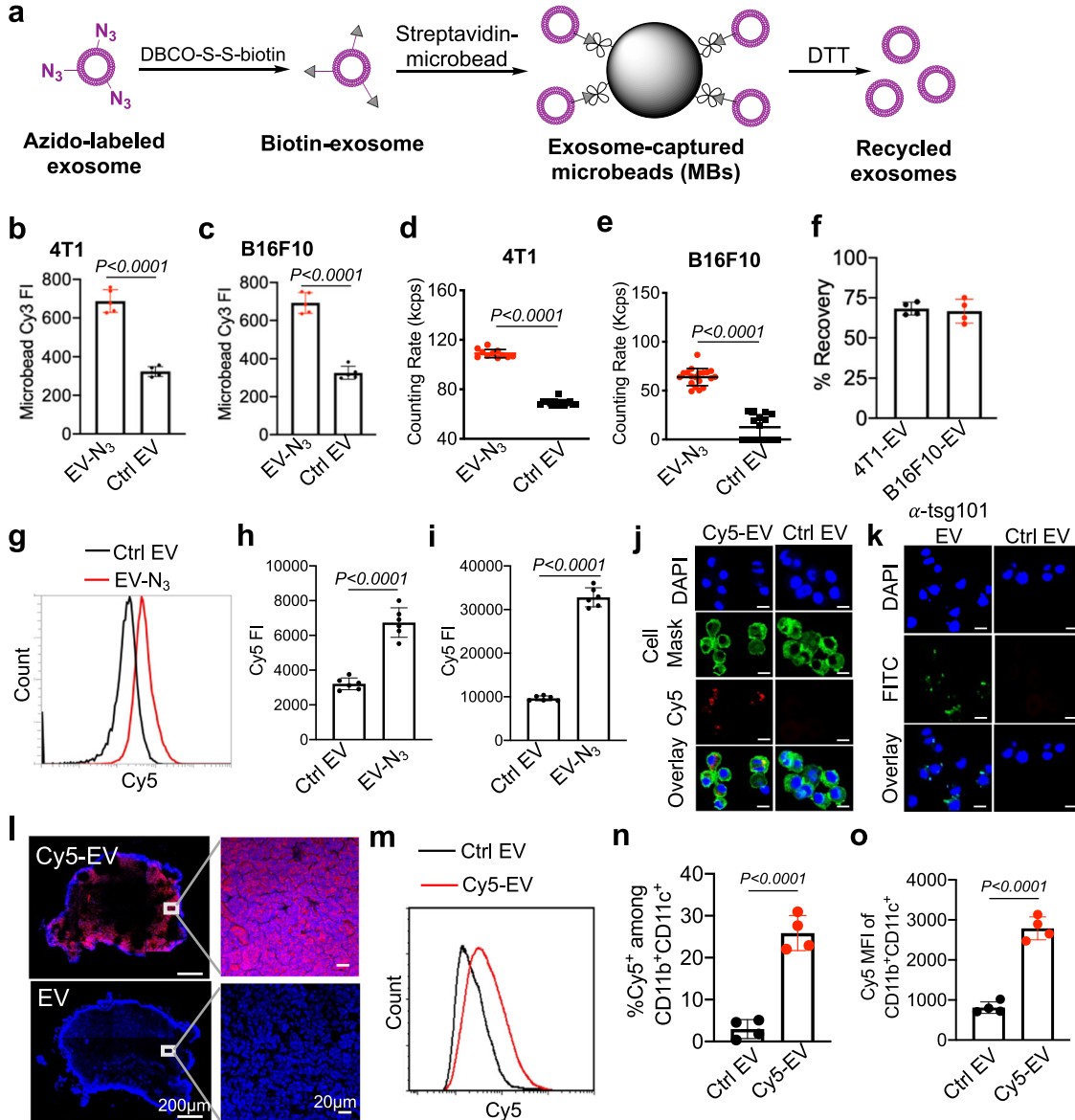

**Fig. 3 | Metabolic tagging of EVs enables isolation and in vitro and in vivo tracking of tumor EVs. a** Schematic illustration of the isolation of azido-labeled tumor EVs. EVs from Ac4ManAz-treated cells can be conjugated with DBCO-S-S-biotin and DBCO-Cy3 and subsequently bounded to streptavidin-modified microbeads. The magnetic microbeads can be collected and treated with DTT to release bounded EVs. Shown are the mean Cy3 FI of microbeads capturing azido-labeled EVs or unlabeled EVs for (**b**) 4T1 cells ($n = 5$) and (**c**) B16F10 cells ($n = 5$). Counting rates of recycled EVs derived from Ac4ManAz-treated (**d**) 4T1 ($n > 10$) and (**e**) B16F10 cells ($n > 20$), respectively. **f** Recovery efficiency of 4T1 or B16F10 tumor EVs ($n = 4$). **g–j** BMDCs were incubated with Cy5-conjugated E.G7-OVA EVs or control EVs for 30 or 60 min. **g** Representative Cy5 histogram of BMDCs after 30-min incubation with EVs. Also shown is the mean Cy5 FI of BMDCs after (**h**) 30-min or (**i**) 60-min incubation with EVs ($n = 6$). **j** CLSM images of BMDCs after 60-min incubation with Cy5-conjugated EVs. Cell nuclei and membranes were stained with DAPI (blue) and Cell Mask Stain (green), respectively. Scale bar: 10 μm. **k** CLSM images of BMDCs after 60-min incubation with anti-tsg101-conjugated EVs. Cell nuclei were stained with DAPI (blue). FITC-conjugated anti-tsg101 (green) was used to stain EVs. Scale bar: 10 μm. **l–o** Cy5-conjugated E.G7-OVA EVs or control EVs were subcutaneously injected into the flank of C57BL/6 mice ($n = 4$), followed by the analysis of draining lymph nodes after 16 h. **l** Confocal image of lymph node sections. Cell nuclei were stained with DAPI (blue). **m** Representative Cy5 histogram of CD11c+ DCs in lymph nodes. **n** Percentage of Cy5+ cells among CD11b+CD11c+ DCs in lymph nodes ($n = 4$). **o** Mean Cy5 FI of CD11b+CD11c+ DCs in lymph nodes ($n = 4$). All the numerical data are presented as mean ± SD (two-tailed Welch's $t$ test was used; $0.01 <*P ≤ 0.05$; $**P ≤ 0.01$; $***P ≤ 0.001$; $****P ≤ 0.0001$). Source data are provided as a Source Data file.

control cells without Ac4ManAz treatment (Fig. S2a, b), demonstrating the successful metabolic labeling of MSCs with azido groups. EVs were then collected from Ac4ManAz-treated or untreated MSCs, and incubated with DBCO-Cy5 for azido detection. Compared to EVs isolated from untreated MSCs, EVs from Ac4ManAz-treated MSCs showed significantly higher Cy5 fluorescence intensity (Fig. S2c), confirming the presence of azido groups on the surface of EVs. Similarly, DCs and T cells can also be metabolically labeled with azido groups by Ac4ManAz (Fig. S2d, e, Fig. S2g, h), resulting in the secretion of azido-tagged EVs (Figs. S2f, S2i). The quantification of Cy5 signal of EVs indicated the presence of >3400, 2300, and 3000 azido groups per EV for MSCs, DCs, and T cells, respectively (Fig. S3b–d). It is noteworthy that metabolic glycan labeling of MSCs, DCs, and T cells did not alter the size and number of secreted EVs (Fig. S4e–h).

**In vitro and in vivo tracking of azido-tagged EVs**

We next studied whether surface azido tags enable the isolation of intact tumor EVs via click chemistry. EVs from Ac4ManAz-treated 4T1

or B16F10 cells were treated with DBCO-S-S-biotin and DBCO-Cy3 to yield biotin/Cy3-conjugated EVs, which were further incubated with streptavidin-modified microbeads to yield EV-microbead conjugates (Fig. 3a). The higher Cy3 fluorescence intensity in the Ac$_4$ManAz treatment group demonstrated the successful capture of azido-tagged EVs by microbeads via azido-DBCO and biotin-streptavidin chemistries (Fig. 3b, c). To further study whether the captured EVs can be intactly released from microbeads, EV-conjugated microbeads were treated with dithiothreitol (DTT) for 10 min to cleave the disulfide bond between EVs and microbeads (Fig. 3a). Dynamic light scattering measurements confirmed the release of EVs from the microbeads (Fig. 3d, e), with a recovery efficiency of 68% and 66% for 4T1 and B16F10 EVs, respectively (Fig. 3f). It is noteworthy that DTT treatment did not disrupt or alter the size of EVs (Fig. S5a–c). In addition to the isolation of EVs, surface azido tags also enable conjugation of DBCO-fluorophores (e.g., DBCO-Cy5) for in vitro and in vivo tracking of EVs. In vitro, upon incubation with bone marrow-derived DCs (BMDCs), flow cytometry analysis revealed the time-dependent cell uptake of Cy5-conjugated EVs (Fig. 3g–i). Confocal imaging confirmed the internalization of Cy5-conjugated EVs by DCs (Fig. 3j). Successful staining of EVs with FITC-conjugated anti-tsg101, a commonly used EV marker, further confirmed the uptake of EVs by DCs (Fig. 3k). In vivo, by subcutaneously injecting Cy5-conjugated EVs into the flank of C57BL/6 mice, a significant amount of Cy5 signal was detected in the draining lymph nodes after 16 h (Fig. 3l), especially within DCs in the lymph nodes (Fig. 3m–o). Cy5-conjugated EVs were also taken up by CD11b$^+$F4/80$^+$ macrophages (Fig. S6a–c), but to a much less extent than DCs (Fig. S6d, e).

## TLR9 agonist-conjugated EVs exhibit superior DC-activating effect

We next conjugated azido-labeled EVs with DBCO-modified CpG, a commonly used TLR9 agonist, and studied whether CpG-conjugated EVs can mediate improved activation of DCs. DBCO-CpG was synthesized by reacting CpG-amine with DBCO-sulfo-NHS, and was incubated with azido-labeled EVs to yield CpG-conjugated EVs (Fig. 4a). To study whether CpG-conjugated, 4T1-derived EVs can improve the activation of DCs, BMDCs were incubated with CpG-conjugated EVs, a mixture of CpG and EVs, CpG, EVs, or PBS for 16 h, with the same final concentration of CpG (1 nM) and EVs ($1 \times 10^7$/mL). Compared to the mixture of CpG and EVs, CpG-conjugated EVs resulted in a significantly higher percentage of CD86$^+$MHCII$^+$ DCs (5.8-fold increase, Fig. 4b) and expression level of CD86 (Fig. 4c). It is noteworthy that the mixture of CpG and EVs did not improve the activation of DCs in comparison with EV alone or CpG alone (Fig. 4b, c). A similar phenomenon, i.e., improved DC activation effect of CpG-conjugated EVs, was observed for E.G7-OVA lymphoma and B16F10 melanoma-derived EVs (Fig. 4d–g, Fig. S7a–c). CpG-conjugated E.G7-OVA-derived EVs showed dramatically improved activation of DCs compared to the mixture of CpG and EVs, with a 4.0-fold and 3.5-fold increase in the percentage of CD86$^+$ DCs and MHCII$^+$ DCs, respectively (Fig. 4d–g). By fixing the concentration of E.G7-OVA derived EVs ($1 \times 10^7$/mL) while increasing the concentration of CpG from 1 nM to 5 nM or 20 nM, a higher level of CD86 and MHCII was consistently observed for CpG-conjugated EVs in comparison with the mixture of CpG and EVs, CpG alone, or EV alone (Fig. 4h–k). We also studied the dose-dependent DC-activating effect by treating BMDCs with varying concentrations of EVs (1, 2, 3, 7, 25, or $70 \times 10^7$/mL) and CpG (1, 2, 3, 7, 25, or 70 nM; concentration ratio of EVs and CpG is fixed). At all doses, CpG-conjugated EVs resulted in significantly higher expression levels of CD86 and MHCII compared to EV alone or the mixture of EV and CpG (Fig. 4l–o, Fig. S8a–l, Fig. S9a-l). Strikingly, CpG-conjugated EVs, at an EV concentration of $1 \times 10^7$/mL and CpG concentration of 1 nM, resulted in dramatically improved activation of DCs in comparison with the mixture of EVs and CpG with an EV concentration of $7 \times 10^8$/mL and CpG concentration of 70 nM

(Fig. 4l–o, Fig. S8a-l, Fig. S9a-l). These experiments demonstrated the superior DC-activating effect of CpG-conjugated EVs.

## TLR9 agonist-conjugated tumor EVs improve antigen presentation by DCs

After demonstrating the superior DC-activating ability of CpG-conjugated EVs, we next studied whether CpG-conjugated E.G7-OVA derived EVs can improve the processing and presentation of EV-encased antigens (e.g., ovalbumin (OVA) CD8 epitope, SIINFEKL) by DCs (Fig. 5a). The presence of OVA protein in E.G7-OVA derived EVs instead of B16F10 derived EVs was confirmed via western blot (Fig. 5b). We also successfully detected the presence of tsg101 and CD63 proteins, two commonly used EV markers, in E.G7-OVA derived EVs (Fig. 5b). BMDCs were then incubated with CpG-conjugated EVs, a mixture of EV and CpG, EV alone, and PBS, respectively for 16 h, followed by the detection of expressed MHCI-SIINFEKL complexes via flow cytometry analysis. CpG-conjugated EVs resulted in a significantly higher expression level of MHCI-SIINFEKL complexes on DCs compared to all the control groups (Fig. 5c, Fig. S10a–c), demonstrating the improved processing and presentation of antigens encased in CpG-conjugated EVs by DCs. It is noteworthy that the mixture of CpG and EVs did not result in any improvement in the presentation of SIINFEKL antigen by DCs (Fig. 5c, Fig. S10a–c), which is consistent with the DC activation results above. DCs with these treatments were further co-cultured with carboxyfluorescein diacetate succinimidyl ester (CFSE)-stained, SIINFEKL-specific OT-1 cells for three days. As expected, DCs pretreated with CpG-conjugated EVs resulted in a significantly improved proliferation of OT-1 cells in comparison with all the control groups (Fig. 5d, e). By lowering the concentration of EVs from $7 \times 10^8$ to $2.5 \times 10^8$ or $7 \times 10^7$ or $1 \times 10^7$/mL, CpG-conjugated EVs consistently resulted in the improved proliferation of OT-1 cells compared to the mixture of CpG and EVs or EV alone (Fig. S11a–c). These experiments substantiated the ability of CpG-conjugated EVs to enhance the presentation of EV-encased antigens by DCs and subsequent priming of antigen-specific CD8$^+$ T cells.

## TLR9 agonist-conjugated tumor EVs show enhanced CTL response

We next studied the CTL response and antitumor efficacy of CpG-conjugated E.G7-OVA EVs. C57BL/6 mice were subcutaneously injected with CpG-conjugated EVs, a mixture of CpG and EV, EV alone, or PBS on days 1, 4, and 7 (Fig. 5f). Peripheral blood mononuclear cells (PBMCs) were then harvested for the analysis of SIINFEKL-specific CD8$^+$ T cells. On day 6, 9, or 12, a significantly higher frequency of SIINFEKL-MHCI tetramer$^+$ CD8$^+$ T cells was detected in mice treated with CpG-conjugated EVs, compared to mice treated with the mixture of CpG and EV or EV alone (Fig. 5g, Fig. S12a–c). Compared to EV alone, the mixture of CpG and EVs resulted in a negligible change in the number of SIINFEKL-specific CD8$^+$ T cells in PBMCs (Fig. 5g, Fig. S12a–c). On day 20, a similar trend, i.e., higher numbers of tetramer$^+$ CD8$^+$ T cells in mice treated with CpG-conjugated EVs than control mice, was observed (Fig. 5h, Fig. S12d). IFN-γ$^+$ CD8$^+$ T cells, after ex vivo SIINFEKL restimulation, also showed a higher frequency in mice treated with CpG-conjugated EVs (Fig. 5i, Fig. S12e). To further amplify SIINFEKL-specific T cell response, a booster dose of EV vaccine was administered into mice on day 32. Three days after the booster, mice treated with CpG-conjugated EVs still showed a significantly higher number of SIINFEKL-specific CD8$^+$ T cells in PBMCs (Fig. 5j–l). In the following prophylactic tumor study, compared to the mixture of tumor EVs and CpG or EV alone, CpG-conjugated EVs resulted in significantly improved tumor control and animal survival (Fig. 5m, n, Fig. S12f). In contrast, the mixture of tumor EVs and CpG failed to exert any benefit compared to EV alone (Fig. 5m, n, Fig. S12f). These experiments substantiated the superior ability of CpG-conjugated tumor EVs to elicit enhanced CTL response and antitumor efficacy.

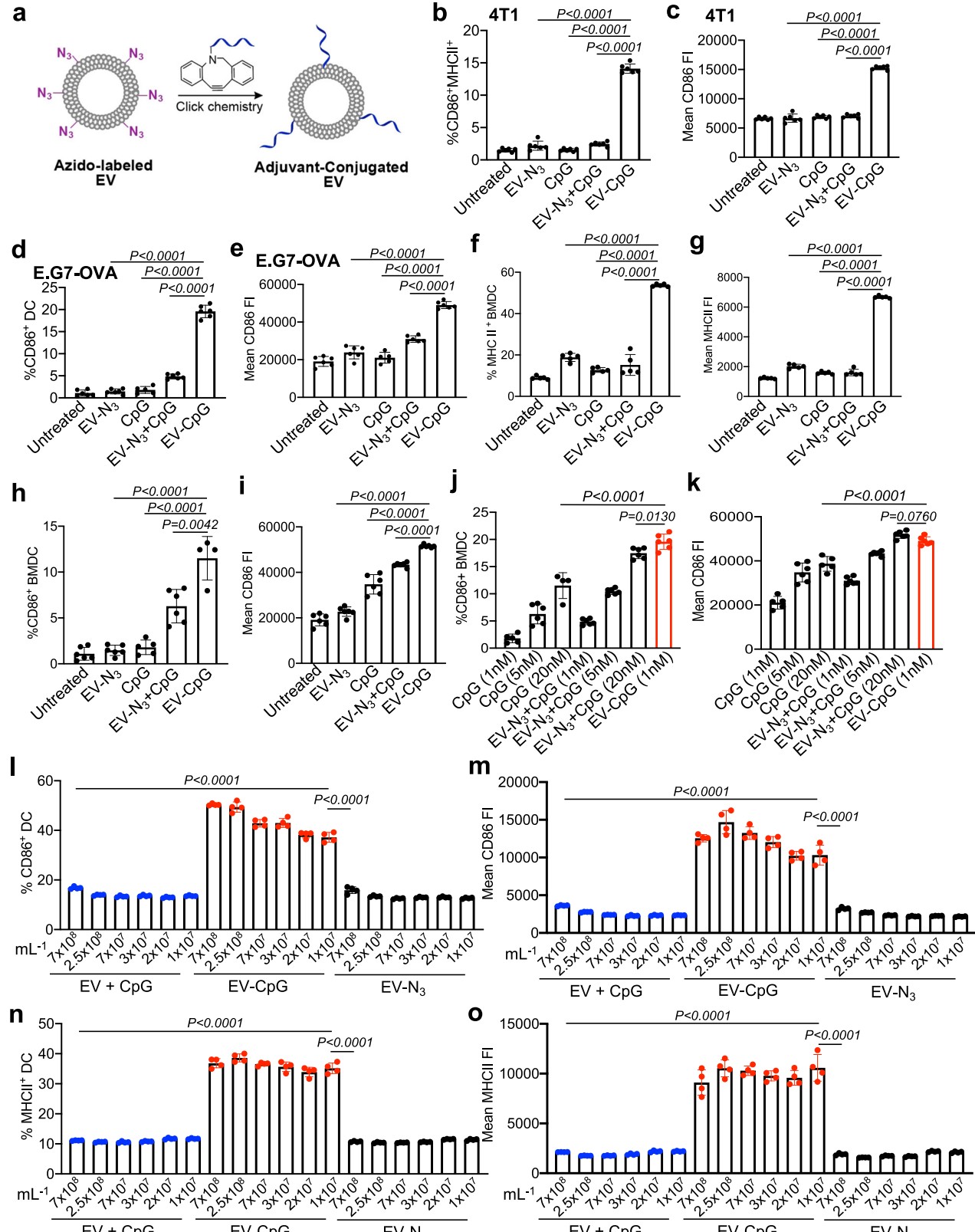

### TLR9 agonist-conjugated tumor EVs show enhanced therapeutic efficacy

In a therapeutic setting, C57BL/6 mice bearing established E.G7-OVA tumors were administered with CpG-conjugated E.G7-OVA-derived EVs, the mixture of CpG and EVs, or EV alone (Fig. 6a). Mice treated with anti-PD-1 or the combination of EVs and anti-PD-1 were also used as controls. All the treatment groups were able to inhibit the growth of

tumors compared to the untreated group (Fig. 6b, c, Fig. S13a, b). Compared to the mixture of CpG and EVs or EV alone, CpG-conjugated EVs further inhibited the tumor growth and prolonged the animal survival (Fig. 6b, c, Fig. S13a, b). Compared to EV alone, the mixture of CpG and EVs failed to exert any improvement in animal survival (Fig. 6b, c). CpG-conjugated EVs also resulted in enhanced antitumor efficacy compared to anti-PD-1 or the combination of anti-PD-1 and EVs

**Fig. 4 | TLR9 agonist-conjugated tumor EVs exhibit superior DC activation.**
**a** Synthesis of CpG-conjugated EVs via conjugation of DBCO-CpG to azido-labeled EVs. For (**b**–**g**), DCs were incubated with CpG (1 nM)-conjugated EVs ($1 \times 10^7$/mL), the mixture of CpG and EVs, EV alone, or CpG alone for 16 h. Shown are (**b**) percentages of CD86⁺MHCII⁺ DCs ($n = 6$) and (**c**) mean CD86 FI of DCs after treatment with 4T1-derived EVs ($n = 6$). **d** Percentage of CD86⁺ DCs and (**e**) mean CD86 FI of DCs after treatment with E.G7-OVA-derived EVs ($n = 6$). **f** Percentage of MHCII⁺ DCs and (**g**) mean MHCII FI of DCs after treatment with E.G7-OVA-derived EVs for 16 h ($n = 5$). **h**–**i** DCs were incubated with CpG (5 nM)-conjugated E.G7-OVA EVs ($1 \times 10^7$/mL), the mixture of CpG and EVs, EV alone, or CpG alone for 16 h ($n = 6$). Shown are (**h**) percentages of CD86⁺ DCs and (**i**) mean CD86 FI of DCs after different treatments.

**j** Percentage of CD86⁺ DCs and (**k**) mean CD86 FI of DCs after incubation with CpG (1 nM)-conjugated EVs ($1 \times 10^7$/mL) or CpG alone (1, 5, or 20 nM) or the mixture of CpG (1, 5, or 20 nM) and EVs ($1 \times 10^7$/mL) for 16 h ($n = 6$). **l**–**o** DCs were treated with CpG-conjugated E.G7-OVA EVs, the mixture of EVs and CpG, or EV alone with varying concentrations of EVs and CpG for 16 h. The concentration ratio of CpG to EVs was fixed (1 nM per $1 \times 10^7$/mL EVs). Shown are (**l**) percentages of CD86⁺ DCs ($n = 4$), (**m**) mean CD86 FI of DCs ($n = 4$), (**n**) percentages of MHCII⁺ DCs ($n = 4$), and (**o**) mean MHCII FI of DCs after different treatments ($n = 4$). All the numerical data in these figures are presented as mean ± SD (one-way ANOVA with post hoc Fisher's LSD test was used; $0.01 < *P \leq 0.05$; $**P \leq 0.01$; $***P \leq 0.001$). Source data are provided as a Source Data file.

(Fig. 6b, c, Fig. S13a, b). It is noteworthy that CpG-conjugated EVs did not exhibit any noticeable toxicity in the examined tissues including spleen, liver, kidney, lung, and heart (Fig. 6d, Fig. S13c).

To expand the applicability of adjuvant-conjugated EV vaccines, we also tested the CTL response and antitumor efficacy of CpG-conjugated B16F10-derived EVs against B16F10 melanoma. We first studied whether subcutaneous injection of CpG-conjugated B16F10-derived EVs could induce the generation of B16F10-specific CD8⁺ T cells in C57BL/6 mice. At 15 or 18 days post vaccination, PBMCs were isolated from mice and restimulated ex vivo with trp2 and gp100 peptides. Compared to the untreated group, CpG-conjugated EVs resulted in a significantly higher number of IFN-γ⁺ CD8⁺ T cells in PBMCs (Fig. S14a–e), demonstrating the ability of CpG-conjugated B16F10-derived EVs to elicit B16F10-specific CD8⁺ T cell response. The combination of CpG-conjugated EVs and anti-PD-1 was able to slightly improve the B16F10-specific CD8⁺ T cell response and the control of B16F10 tumor growth in the following prophylactic tumor study (Fig. S14f). In a therapeutic setting, different from the E.G7-OVA model, the mixture of CpG and EVs or EV alone failed to exert any therapeutic benefit against B16F10 melanoma, in comparison with the untreated group (Fig. 6e–g, Fig. S15a, b). Mice treated with anti-PD-1 or the combination of anti-PD-1 and EVs also showed similar tumor growth rates and animal survival to the untreated mice (Fig. 6f, g, Fig. S15a, b). Among them, CpG-conjugated EVs exhibited the best antitumor efficacy with a reduced tumor growth rate and prolonged animal survival (Fig. 6f, g, Fig. S15a, b), without inducing any sign of toxicity towards the healthy tissues (Fig. 6h, Fig. S15c).

To better understand the alteration of the tumor microenvironment as a result of EV vaccination, we also analyzed the number, activation status, and memory phenotype of T cells in the tumors and tumor-draining lymph nodes at 6 days post vaccination (Fig. S16a). Compared to EV alone or the mixture of EVs and CpG, CpG-conjugated EVs resulted in a higher number of CD8⁺ T cells (Fig. S16b, c), a lower number of regulatory T (Treg) cells (Fig. S16d), and a higher CD8⁺/Treg ratio (Fig. S16e). The intratumoral CD8⁺ T cells also exhibited an upregulated expression of CD69 activation marker (Fig. S16f) and increased fraction of central memory phenotype (CD44⁺CD62L⁺) and effector memory phenotype (CD44⁺CD62L⁻) in mice treated with CpG-conjugated EVs, compared to mice treated with EV alone or the mixture of EVs and CpG (Fig. S16g, h). It is noteworthy that CpG-conjugated EVs also resulted in an increased expression of PD-1 by intratumoral CD8⁺ T cells than the control groups (Fig. S16i), which is consistent with previous reports that activated CD8⁺ T cells tend to express a higher level of exhaustion markers[38–41]. In the tumor-draining lymph nodes, mice treated with CpG-conjugated EVs also showed a higher CD8/Treg ratio and an increased number of CD69⁺ CD8⁺ T cells and memory-phenotype CD8⁺ T cells, in comparison with mice treated with EV alone or the mixture of EV and CpG (Fig. S17a–e). These experiments demonstrated the superior antitumor efficacy of CpG-conjugated tumor EVs over conventional EV vaccines, either EV alone or the mixture of adjuvants and EVs.

## Discussion

EVs have been actively explored for diagnostic and therapeutic applications, but approaches to track and target EVs are still lacking. Our metabolic labeling approach can introduce >3000 clickable chemical tags onto each EV, for subsequent tracking and targeted modulation of EVs. This approach is universally applicable to various types of cells including cancer cells, MSCs, DCs, and T cells. Metabolic glycan labeling of cells can introduce up to $10^8$–$10^9$ azido groups onto the cell membrane, which is 3–4 orders of magnitude higher than conventional protein receptors (up to $10^5$ per cell)[15,16]. Considering the 10,000-fold surface area difference between a 10-μm cell and a 100-nm EV, in principle, up to $10^4$–$10^5$ chemical tags can be inherited by each cell-secreted EV. In addition to azido groups, the EV tagging technology can also introduce other types of chemical tags such as ketone, alkyne, alkene, and norbornene groups onto the surface of EVs[15,42,43]. This will allow one to tag and distinguish EVs secreted by different types of cells, facilitating the understanding of EV-mediated intercellular communication.

The surface azido groups enable the quantitative conjugation of DBCO-molecules via efficient click chemistry, with a tunable density of conjugated molecules. This enables the conjugation of a good amount of fluorophores or contrast agents for in vivo tracking of EVs, immunomodulatory agents to regulate the immunogenicity of EVs, and other molecules to modulate the interaction between EVs and recipient cells. The easiness of displaying a tunable amount of molecule of interest to EVs, together with the universality and simplicity of the metabolic tagging technology, is expected to greatly facilitate future EV research and the development of EV-based diagnostics and therapeutics.

Compared to killed tumor cells or tumor cell lysates, tumor EVs are regarded as a safer source of tumor antigens for developing therapeutic vaccines[38,44–46]. Indeed, tumor EV vaccines have shown the promise to induce antitumor CTL response in clinical trials, albeit with a modest therapeutic efficacy[28,29]. The incorporation of adjuvants (e.g., alum and CpG) is a common strategy to amplify the immunogenicity of conventional vaccines[47–49]. However, the mixture of tumor EVs and adjuvants could barely improve the CTL response and antitumor efficacy of tumor EV vaccines[35,50,51], which is also supported by our data where the mixture of CpG and tumor EVs failed to exert any noticeable improvement in CTL response and therapeutic efficacy compared to EV alone (Figs. 5–6). In contrast, the conjugation of sufficient CpG molecules to EVs, using our EV tagging and targeting approach, managed to dramatically improve the activation of DCs in comparison with CpG or the mixture of CpG and EVs (Fig. 4). This phenomenon, i.e., enhanced DC activation of CpG-conjugated EVs, was observed across various types of EVs including 4T1 breast cancer EV, E.G7-OVA lymphoma EV, and B16F10 melanoma EV (Fig. 4, Fig. S7). One of our hypotheses is that the metabolic tagging approach can install a high density of TLR9 agonist (CpG) on the surface of EVs. Once the CpG-conjugated EVs are endocytosed by DCs via endosomes, the nanocluster of CpG on the surface of EVs can effectively stimulate the TLR9 scattered on the surface of endosomes, resulting in the timely activation of DCs. We further show that CpG-conjugated EVs can significantly improve the processing and presentation of EV-encased antigens by DCs (Fig. 5a–e), which could be attributed to the timely

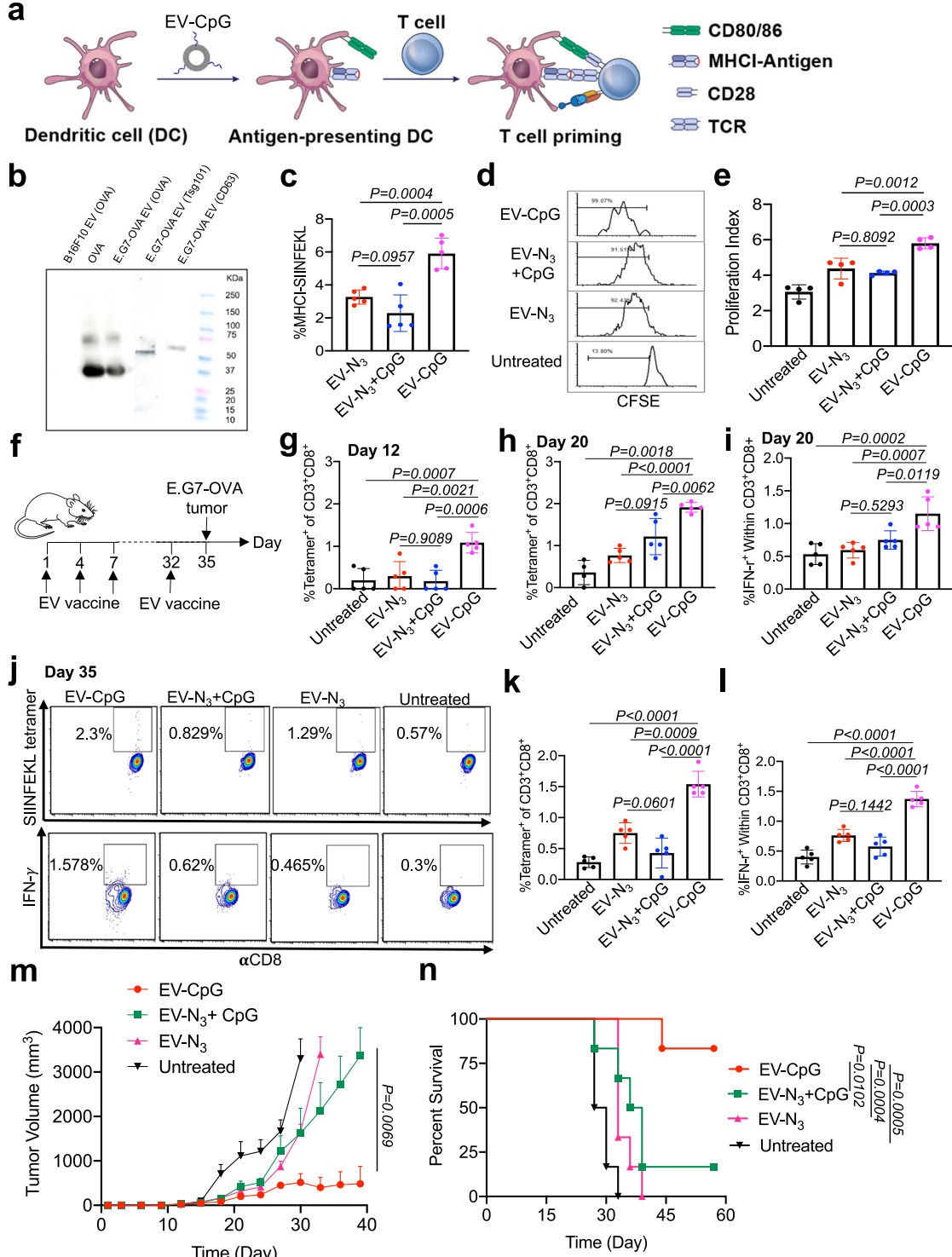

stimulation of TLR9 on endosomes during the endocytosis of tumor EVs.

Vaccination of mice with CpG-conjugated tumor EVs was able to elicit an enhanced CTL response than the mixture of CpG and EVs or EV alone (Fig. 5). In therapeutic settings, CpG-conjugated EVs also resulted in significantly improved antitumor efficacy against both E.G7-OVA lymphoma and B16F10 melanoma, in comparison with the mixture of CpG and EVs, EV alone, anti-PD-1, or the combination of anti-PD-1 and EVs (Fig. 6). While exhibiting enhanced CTL response and antitumor efficacy, CpG-conjugated EVs did not result in increased toxicity compared to EV alone or the mixture of EVs and CpG (Fig. 6d, h).

In addition to lymphoma and melanoma, we envision our adjuvant conjugation approach can improve the CTL response and therapeutic efficacy of EV vaccines against other types of tumors[39–41]. The ability to manipulate the interaction between endosomes and intracellular EVs also hold the promise to improve the efficacy of EV vaccines, and more broadly EV-based therapies, in the context of non-cancer diseases.

To conclude, we report an EV tagging technology that enables the generation of chemically tagged EVs from various types of cells, and the development of next-generation EV vaccines with superior T cell response and therapeutic efficacy. We demonstrated that metabolic glycan labeling of cells with azido-sugars resulted in the secretion of

**Fig. 5 | TLR9 agonist-conjugated tumor EVs result in improved processing and presentation of EV-encased antigens by DCs and enhanced CTL response.**
**a** Schematic illustration for the presentation of EV-encased antigens by DCs and subsequent priming of antigen-specific CD8[+] T cells. **b** Western blot analysis of OVA protein from E.G7-OVA EVs, B16F10 EVs, or pure OVA control. Also shown are the analysis of Tsg101 and CD63 proteins, two commonly used EV markers, from E.G7-OVA EVs. This experiment was repeated at least once with similar results.
**c** Percentage of MHCI-SIINFEKL[+] DCs after incubation with CpG (1 nM)-conjugated EVs ($1 \times 10^7$/mL), EVs alone, or the mixture of CpG and EVs for 16 h ($n = 5$). **d–e** DCs pretreated with CpG-conjugated EVs and other control groups (70 nM CpG and $7 \times 10^8$/mL EVs) were cocultured with CFSE-stained OT-1 cell for three days. Shown are (**d**) representative CFSE histograms and (**e**) proliferation index of OT-1 cells in different groups ($n = 4$). **f** Timeframe of the vaccination study ($n = 6$ per group). CpG-conjugated EVs, the mixture of CpG and EVs, EVs alone, or PBS were

subcutaneously injected into C57BL/6 mice on day 1, 4, 7 and 32. E.G7-OVA tumor cells were inoculated on day 35. Shown are the percentage of SIINFEKL tetramer[+] cells among CD8[+] T cells in PBMC on (**g**) day 12 and (**h**) day 20, respectively ($n = 5$). **i** Percentage of IFN-γ[+] cells among CD8[+] T cells in PBMC on day 20, upon ex vivo restimulation with SIINFEKL peptide. **j** Representative FACS plots of tetramer[+] CD8[+] T cells and IFN-γ[+] CD8[+] T cells in PBMC on day 35. **k** Percentage of tetramer[+] cells among CD8[+] T cells in PBMC on day 35 ($n = 5$). **l** Percentage of IFN-γ[+] cells among CD8[+] T cells in PBMC on day 35 ($n = 5$). **m** Average E.G7-OVA tumor volume of each group over the course of the prophylactic tumor study ($n = 6$). **n** Kaplan-Meier plots for all groups ($n = 6$). All the numerical data are presented as mean ± SD except for (**m**) where data are presented as mean ± SEM (one-way ANOVA with post hoc Fisher's LSD test was used; $0.01 < *P \le 0.05$; $**P \le 0.01$; $***P \le 0.001$). Source data are provided as a Source Data file.

azido-labeled EVs, which enabled the conjugation of DBCO-molecules via efficient click chemistry for in vitro and in vivo tracking and targeted modulation of EVs. In the context of tumor EVs, we showed that TLR9 agonist-conjugated EVs could dramatically improve the activation of DCs, processing and presentation of EV-encased antigens by DCs, and subsequent priming of antigen-specific CD8[+] T cells, leading to significantly improved CTL response and antitumor efficacy against lymphoma and melanoma. The EV tagging and targeting technology will greatly facilitate the development of EV-based therapies, and provides a facile platform to manipulate EV-cell interactions and decipher the role of EVs in different diseases.

## Methods

### Ethical statement
This research complies with all relevant ethical regulations. All procedures involving animals were done in compliance with National Institutes of Health and Institutional guidelines with approval from the Institutional Animal Care and Use Committee at the University of Illinois at Urbana-Champaign.

### Materials and instrumentation
D-Mannosamine hydrochloride, DBCO-Cy5, DBCO-Cy3, sodium azide, bromoacetic acid, dicyclohexyl carbodiimide, *N*-hydroxysuccinimide, and other chemical reagents are purchased from Sigma Aldrich (St. Louis, MO, USA) unless otherwise noted. DBCO-S-S-Biotin was purchased from Click Chemistry Tools (Scottsdale, AZ, USA). Streptavidin microbeads were purchased from Thermo Fisher Scientific (Waltham, MA, USA). Recombinant murine GM-CSF was purchased from Pepro-Tech, Inc. (Cranbury, NJ, USA). Primary antibodies used in this study include fluorophore-conjugated anti-CD11b (Invitrogen, Catalog #14-0112-82), anti-CD11c (Invitrogen, Catalog #12-0114-82), anti-CD86 (Invitrogen, Catalog #12-0862-82), anti-MHCII (Invitrogen, Catalog #11-0920-82), anti-CD3 (Invitrogen, Catalog #17-0031-82), anti-CD8 (Invitrogen, Catalog #45-0081-82), anti-CD4 (Invitrogen, Catalog #12-0041-82), anti-F4/80 (Invitrogen, Catalog #17-4801-82), anti-MHCI-SIINFEKL (Invitrogen, Catalog #12-5743-82), and anti-IFN-γ (Invitrogen, Catalog #17-7311-82). Fixable viability dye efluor780 (Catalog #65-0865-14) was obtained from Thermo Fisher Scientific. All antibodies were diluted 100 times, except for the fixability dye which was diluted 1000 times. SIINFEKL-MHCI tetramer was requested from the NIH Tetramer Core. HRP-conjugated OVA polyclonal antibody (Catalog #PA1-196-HRP) was purchased from Thermo Fisher Scientific. Mouse CD3[+] T cell isolation kit, dynabeads, and LS separation columns were purchased from Miltenyi Biotec (Bergisch Gladbach, Germany). qEV isolation columns were purchased from IZON Science (Christchurch, New Zealand). FACS analyses were collected on Attune NxT or BD LSR Fortessa flow cytometers and analyzed on FlowJo v7.6 and FCS Express v6 and v7. Statistical testing was performed using GraphPad Prism v6 and v8. Fluorescence measurement of DBCO-Cy3 and DBCO-Cy5 was conducted on a plate reader. Small compounds were run on the

Agilent 1290/6140 ultra high-performance liquid chromatography/ mass spectrometer. Proton nuclear magnetic resonance spectra were collected on the Agilent DD2 600. Matrix-assisted laser desorption/ ionization mass spectra were collected on the Bruker Ultraflextreme MALDI-TOF/TOF Mass Spectrometer. The size and size distribution of EVs were measured on a dynamic light scattering (DLS) instrument and Nanoparticle Tracking Analysis (NTA) instrument. Transmission electron microscopic images of EVs were taken with a JEOL 2100 TEM.

### Cell line and animals
The 4T1 (ATCC CRL-2539), LS174T (ATCC CL-188), BxPC-3 (ATCC CRL-1687), B16F10 (ATCC CRL-6475), and E.G7-OVA (ATCC CRL-2113) cell lines were purchased from American Type Culture Collection (Manassas, VA, USA). Cells were cultured in DMEM containing 10% FBS, 100 units/mL Penicillin G and 100 μg/mL streptomycin at 37 °C in 5% $CO_2$ humidified air. E.G7-OVA cells were cultured in the presence of G418. Female C57BL/6 mice were purchased from the Jackson Laboratory (Bar Harbor, ME, USA). Feed and water were available ad libitum. Artificial light was provided in a 12 h/12 h cycle. All procedures involving animals were done in compliance with National Institutes of Health and Institutional guidelines with approval from the Institutional Animal Care and Use Committee (IACUC) at the University of Illinois at Urbana-Champaign (IACUC protocol number 20183).

### Synthesis of Ac4ManAz
D-Mannosamine hydrochloride (1.0 mmol) and triethylamine (1.0 mmol) were dissolved in methanol, followed by the addition of *N*-(2-azidoacetyl) succinimide (1.2 mmol). The mixture was stirred at room temperature for 24 h. The solvent was removed under reduced pressure and the residue was re-dissolved in pyridine. Acetic anhydride was added, and the reaction mixture was stirred at room temperature for another 24 h. After removal of the solvent, the crude product was purified by silica gel column chromatography using ethyl acetate/ hexane (1/1, v/v) as the eluent to yield a white solid (1/1 α/β isomers). [1]H NMR (CDCl$_3$, 500 MHz): δ (ppm) 6.66&6.60 (d, $J = 9.0$ Hz, 1H, C(O) N*H*CH), 6.04&6.04 (d, 1H, $J = 1.9$ Hz, NHCHC*H*O), 5.32−5.35&5.04−5.07 (dd, $J = 10.2$, 4.2 Hz, 1H, CH$_2$CHC*H*CH), 5.22&5.16 (t, $J = 9.9$ Hz, 1H, CH$_2$CHCHC*H*), 4.60-4.63&4.71-4.74 (m, 1H, NHC*H*CHO), 4.10-4.27 (m, 2H, C*H*$_2$CHCHCH), 4.07 (m, 2H, C(O)C*H*$_2$N$_3$), 3.80-4.04 (m, 1H, CH$_2$C*H*CHCH), 2.00-2.18 (s, 12H, C*H*$_3$C(O)). [13]C NMR (CDCl$_3$, 500 MHz): δ (ppm) 170.7, 170.4, 170.3, 169.8, 168.6, 168.3, 167.5, 166.9, 91.5, 90.5, 73.6, 71.7, 70.5, 69.1, 65.3, 65.1, 62.0, 61.9, 52.8, 52.6, 49.9, 49.5, 21.1, 21.0, 21.0, 20.9, 20.9, 20.9, 20.8. ESI MS (*m/z*): calculated for C$_{16}$H$_{22}$N$_4$O$_{10}$Na [M+Na]$^+$ 453.1, found 453.1.

### Confocal imaging of metabolically labeled cells
Cancer cells were seeded onto coverslips in a 6-well plate at a density of $4 \times 10^4$ cells per well and allowed to attach for 12 h. Ac4ManAz (50 μM) was added, and the cells were incubated at 37 °C for 72 h. After washing with PBS, cells were incubated with DBCO-Cy5 (25 μM) for

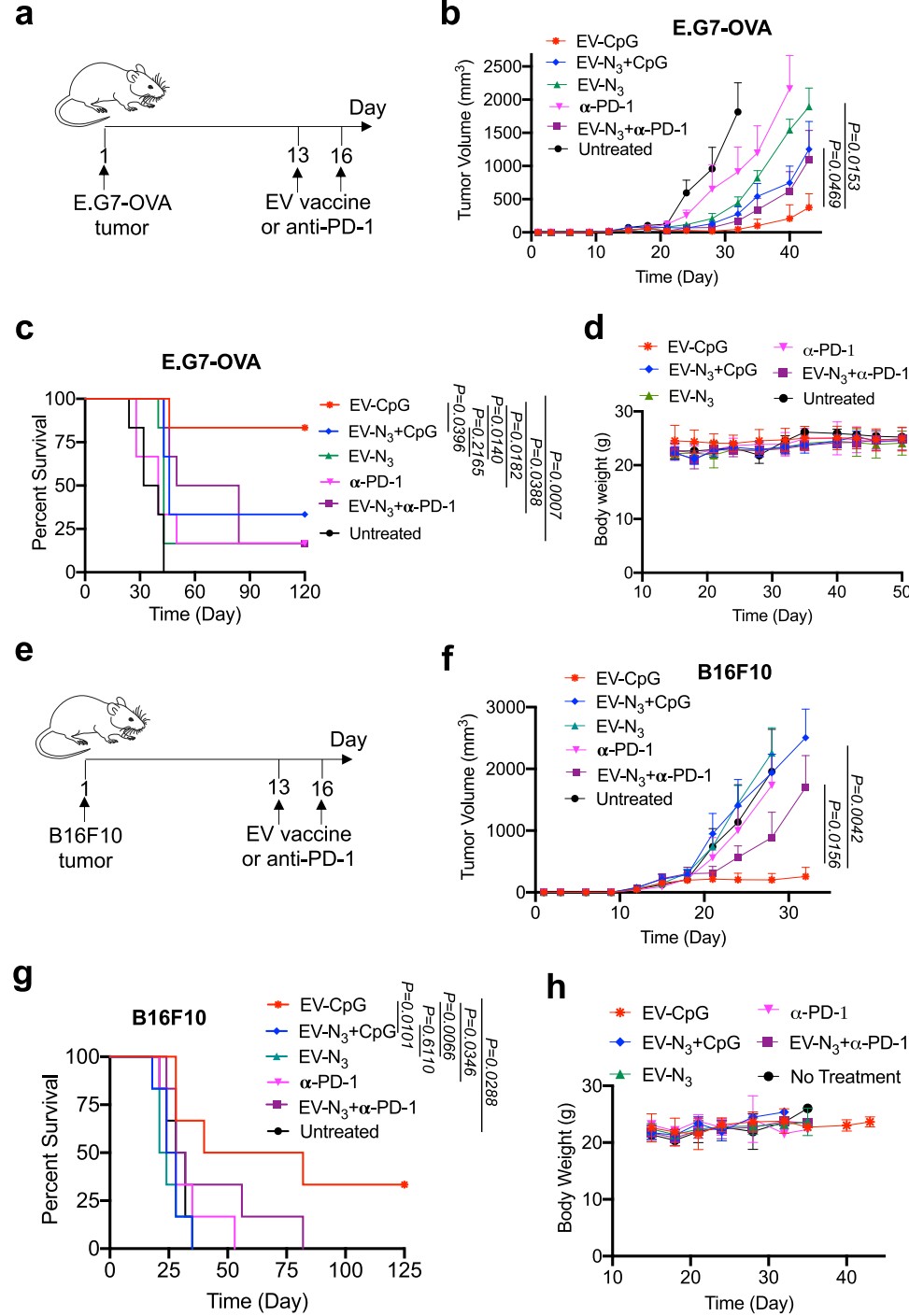

**Fig. 6 | TLR9 agonist-conjugated tumor EVs exhibit enhanced therapeutic efficacy against E.G7-OVA lymphoma and B16F10 melanoma. a** Timeframe of therapeutic tumor study. E.G7-OVA tumor was inoculated on day 0 ($n = 6$ per group). CpG-conjugated EVs, the mixture of CpG and EVs, or EV alone were sub-cutaneously injected on days 13 and 16. Anti-PD-1 was i.p. administered on days 13 and 16. **b** Average E.G7-OVA tumor volume of each group over the course of the therapeutic tumor study ($n = 6$). **c** Kaplan-Meier plots for all groups ($n = 6$). **d** Body weight of mice over the course of efficacy study. **e** Timeframe of therapeutic tumor study. B16F10 tumor was inoculated on day 0 ($n = 6$ per group). CpG-conjugated

EVs, the mixture of CpG and EVs, or EV alone were subcutaneously injected on days 13 and 16. Anti-PD-1 was i.p. administered on days 13 and 16. **f** Average B16F10 tumor volume of each group over the course of the therapeutic tumor study ($n = 6$). **g** Kaplan–Meier plots for all groups ($n = 6$). **h** Body weight of mice over the course of efficacy study. All the numerical data are presented as mean ± SD except for (**b**) and (**f**) where data are presented as mean ± SEM (one-way ANOVA with post hoc Fisher's LSD test was used; $0.01 < *P ≤ 0.05$; $**P ≤ 0.01$; $***P ≤ 0.001$). Source data are provided as a Source Data file.

30 min and fixed with 4% paraformaldehyde solution, followed by staining of cell nuclei and membrane with DAPI. The coverslips were mounted onto microscope slides and imaged under a confocal laser scanning microscope.

**Flow cytometry analysis of metabolically labeled cells**

Cancer cells were seeded in a 24-well plate at a density of $1 \times 10^4$ cells per well and allowed to attach for 12 h. Ac$_4$ManAz (50 μM) was added and incubated with cells for 72 h. After washing with PBS, cells were

incubated with DBCO-Cy5 (10 μM) for 30 min. Cells were further washed and analyzed by flow cytometry.

## Isolation of EVs

Cells were cultured in T75 or T175 flasks in the presence or absence of Ac$_4$ManAz (50 μM) for 3–4 days. Cell culture medium containing the secreted EVs was collected and concentrated via ultracentrifugation with an Amicon centrifugal filter (100 kDa). EVs were washed with PBS three times and resuspended in PBS. To further purify EVs, a solution of EVs was passed through the qEV size exclusion column. The size and size distribution of EVs were measured on dynamic light scattering, while the absolute concentration of EVs was determined on a Nano-particle Tracking Analysis instrument.

## TEM imaging of EVs

Isolated EVs were added onto formvar/carbon-coated TEM grids (Ted Pella, Redding, CA), allowed to dry, negatively stained with 2% aqueous uranyl acetate, and imaged with a JEOL 2100 TEM at 200 kV.

## Western blot analysis of EVs

EVs were harvested from E.G7-OVA cell culture medium in the presence or absence of Ac$_4$ManAz and purified via ultracentrifugation and qEV column. The purified EVs or cells (as positive controls) were lysed and quantified for protein content via a BCA assay kit (Sigma, USA). Laemmli sample buffer was added to the lysates, followed by boiling the samples at 100 °C for 5 min. 10 μg of proteins were loaded and run on a 12% acrylamide gel (Tris-Glycine/SDS running buffer). Protein bands were then transferred to the PVDF membrane (transfer buffer: 25 mM Tris base, 190 mM glycine, and 20% methanol), stained with HRP-conjugated OVA polyclonal antibody. EVs collected from B16-F10 cells were used as the negative control. EV samples were also stained with HRP-conjugated secondary antibody for Tsg101 and CD63, and imaged via the chemiluminescence method.

## Conjugation of DBCO-Cy5 or DBCO-Cy3 to tumor-derived EVs

EVs collected from the culture media of Ac$_4$ManAz-treated or control cancer cells were incubated with DBCO-Cy5 or DBCO-Cy3 (5 μM) for 30 min. EV solutions were ultra-centrifuged with an Amicon centrifugal filter (100 kDa) to remove the unconjugated or unbounded dye. After three washing steps, EVs were resuspended in PBS for use or storage at 4 °C.

## Uptake of tumor EVs by dendritic cells

BMDCs were differentiated from bone marrow cells following a reported protocol[30,52]. In brief, bone marrow cells were extracted from the tibia and femur of C57BL/6 mice and cultured in RPMI medium containing 20 ng/mL GM-CSF for 7 days. The medium was then replaced with fresh medium containing 10 ng/mL GM-CSF. Cy5 or Cy3-conjugated EVs or control EVs were cultured with BMDCs for 0.5 or 2 h, prior to flow cytometry analysis or confocal imaging.

## Recycling and purification of tumor EVs

Azido-labeled EVs or control EVs were incubated with DBCO-S-S-biotin and DBCO-Cy3 for 30 min. After removal of the residual DBCO-molecules via ultracentrifugation (3k Da cut-off molecular weight), EVs were further incubated with streptavidin-modified microbeads for 30 min. Microbeads were collected via low-speed (350 g) centrifugation. To cleave the disulfide bond between EVs and microbeads, EV-capturing microbeads were treated with dithiothreitol (10 mM) for 10 min, followed by the removal of microbeads and collection of EVs.

## Synthesis of DBCO-CpG

CpG-amine (20 nM) and DBCO-sulfo-NHS (22 nM) were mixed in PBS, and shaken at 4 °C for 24 h. The reaction was monitored via HPLC.

Upon the completion of the reaction, DBCO-CpG was purified via ultracentrifugation (3 kDa cut-off molecular weight) and stored at 4 °C for use.

## Conjugation of DBCO-CpG to azido-labeled EVs

EVs isolated from Ac$_4$ManAz-treated cancer cells or untreated cancer cells were mixed with DBCO-CpG at 4 °C for 4 h to enable thorough conjugation. Unreacted DBCO-CpG was removed via ultracentrifugation (100 kDa molecular weight cutoff). EVs were then collected and stored at 4 °C until use.

## In vitro activation of dendritic cells by CpG-conjugated EVs

Day-7 BMDCs (50 k) in 100 μL of medium were incubated with CpG-conjugated EVs, the mixture of EVs and CpG, EV alone, CpG alone, or PBS for 16 h. For most experiments, the concentrations of CpG and EVs were set at 1 nM and $1 \times 10^7$/mL, respectively. To study the effect of CpG concentration, in some experiments, the concentrations of CpG and EVs were set at 5 or 20 nM and $1 \times 10^7$/mL, respectively. DCs were then stained with fluorophore-conjugated anti-CD11c, anti-CD86, and live/dead stain for 20 min at 4 °C, prior to flow cytometry analysis. In some experiments involving E.G7-OVA-derived EVs, cells were also stained with APC-conjugated anti-MHCI-SIINFEKL, prior to FACS analysis. To further evaluate the dose effect of EVs, in a separate experiment, the concentration of EVs was varied from $1 \times 10^7$, $2 \times 10^7$, $7 \times 10^7$, $2.5 \times 10^8$, to $7 \times 10^8$/mL, while the concentration of CpG increased with the concentration of EVs (1 nM CpG per $1 \times 10^7$/mL EVs, i.e., 70 nM for $7 \times 10^8$/mL EVs).

## Co-culture of dendritic cells and OT-1 cells

After 24-h incubation with CpG-conjugated E.G7-OVA EVs, the mixture of EVs and CpG, EV alone, or PBS, dendritic cells were co-cultured with CFSE-stained OT-1 cells for three days, followed by FACS assay to determine the proliferation index of OT-1 cells[53,54]. For DC pretreatment, the concentration of EVs was varied from $1 \times 10^7$, $7 \times 10^7$, $2.5 \times 10^8$, to $7 \times 10^8$/mL, while the concentration of CpG increased with the concentration of EVs (1 nM CpG per $1 \times 10^7$/mL EVs, i.e., 70 nM for $7 \times 10^8$/mL EVs).

## In vivo tracking of tumor EVs

Cy5-conjugated E.G7-OVA-derived EVs or control EVs were subcutaneously injected into the flank of C57BL/6 mice. After 16 h, lymph nodes were isolated for analysis. For flow cytometry analysis, single cell suspensions from lymph nodes were stained with fluorophore-conjugated anti-CD11b, anti-CD11c, anti-F4/80, and live/dead stain for 20 min. For confocal imaging, lymph nodes were frozen in O.C.T. compound, sectioned into 8 μm slices, and fixed with 4% paraformaldehyde. After washing with PBS, tissue sections were stained with DAPI at 4 °C for 10 min and imaged with a Carl Zeiss LSM 700 confocal microscope.

## Analysis of tumor microenvironment

Female C57BL/6 mice (5–7 weeks) were divided into four groups: EV-N$_3$, EV-N$_3$ + CpG, Ex-CpG, and untreated ($n = 5$ per group). B16F10 cells ($2.5 \times 10^5$ cells in 50 μL of HBSS) were subcutaneously injected into the upper right flank of C57BL/6 mice on day 0. On day 11, CpG-conjugated B16F10-derived EVs, the mixture of EVs and CpG, EV alone, or PBS were subcutaneously injected. On day 17, tumors and tumor-draining lymph nodes were harvested for immune analysis. Tumors and lymph nodes were disrupted using a syringe plunger to release cells. Cells were collected, washed, and stained for flow cytometry analysis. For the evaluation of T cells in the tumor microenvironment or lymph nodes, cells were stained with fluorophore-conjugated anti-CD3, antiCD4, anti-CD8, anti-CD69, anti-FoxP3, anti-PD-1, anti-CTLA-4, anti-LAG-3, anti-TIM-3, anti-CD44, and anti-CD62L in separate panels.

## Vaccination and prophylactic tumor study of E.G7-OVA EV vaccines

Female C57BL/6 mice (5–7 weeks) were divided into 4 groups: EV-CpG, EV-N$_3$ + CpG, EV-N$_3$, untreated ($n = 6$ per group). Mice were subcutaneously injected with CpG-conjugated E.G7-OVA derived EVs, the mixture of EVs and CpG, EV alone, or PBS on days 1, 4, and 7, respectively. Blood was drawn on days 6, 9, 12, and 20, respectively for analysis of SIINFEKL-specific CD8$^+$ T cells via tetramer stain or IFN-γ restimulation. For tetramer analysis, PBMCs were stained with APC-conjugated SIINFEKL tetramer, FITC-conjugated anti-CD3, PE-conjugated anti-CD8, and e780 fixable viability dye for 20 min prior to FACS assay. For IFN-γ restimulation, PBMCs were stimulated with SIINFEKL peptide for 1.5 h, treated with Golgi plug for 2.5 h, stained with FITC-conjugated anti-CD3, PE-conjugated anti-CD8, and e780 fixable viability dye, treated with the fixation & permeabilization buffer, and stained with APC-conjugated anti-IFN-γ, prior to FACS assay. On Day 32, a booster vaccine was administered. In the following prophylactic tumor study, E.G7-OVA tumor cells (0.1 million cells in 50 μL of HBSS) were subcutaneously injected into the upper flank of C57BL/6 mice. The tumor volume and body weight of mice were measured every 3 days. The tumor volume was calculated using the formula (length) × (width)$^2$/2, where the long axis diameter was regarded as the length and the short axis diameter was regarded as the width. Mice were euthanized when the largest diameter of tumors reaches 20 mm or mice became moribund.

## Vaccination and prophylactic tumor study of B16F10 EV vaccines

Female C57BL/6 mice (5–7 weeks) were divided into 4 groups: EV-CpG, EV-CpG + anti-PD-1, anti-PD-1, or untreated ($n = 6$ per group). Mice were subcutaneously injected with CpG-conjugated B16F10-derived EVs on day 1. Anti-PD-1 (100 μg) was intraperitoneally injected on day 1. Blood was drawn on days 15 and 18, respectively for analysis of trp2/gp100-specific CD8$^+$ T cells via IFN-γ restimulation. PBMCs were stimulated with the mixture of trp2 and gp100 peptides for 1.5 h, treated with Golgi plug for 2.5 h, stained with FITC-conjugated anti-CD3, PE-conjugated anti-CD8, and e780 fixable viability dye, treated with the fixation & permeabilization buffer, and stained with APC-conjugated anti-IFN-γ, prior to FACS assay. On Day 32, a booster vaccine was administered. In the following prophylactic tumor study, B16F10 tumor cells (0.1 million cells in 50 μL of HBSS) were subcutaneously injected into the upper flank of C57BL/6 mice. The tumor volume and body weight of mice were measured every 3 days. The tumor volume was calculated using the formula (length) × (width)$^2$/2, where the long axis diameter was regarded as the length and the short axis diameter was regarded as the width. Mice were euthanized when the largest diameter of tumors reaches 20 mm or mice became moribund.

## Therapeutic tumor study of EV vaccines

Female E.G7-OVA or B16F10 tumors (5–7 weeks) were established in C57BL/6 mice by subcutaneous injection of E.G7-OVA cells (5 × 10$^5$ in 50 μL of HBSS) or B16F10 cells (2.5 × 10$^5$ in 50 μL of HBSS) into the right flank. When the tumors reached a diameter of 6–7 mm, mice were randomly divided into 6 groups: EV-CpG, Exo-N$_3$ + CpG, Exo-N$_3$, α-PD-1+Exo-N$_3$, anti-PD-1, or untreated. Mice were subcutaneously injected with CpG-conjugated EVs (7 × 10$^8$ EVs and 445 ng CpG), the mixture of EVs and CpG (7 × 10$^8$ EVs and 445 ng CpG), or EV alone (7 × 10$^8$ EVs) on days 13 and 16. Anti-PD-1 (100 μg) was intraperitoneally injected on Days 13 and 16. The tumor volume and body weight of mice were measured every other day. The tumor volume was calculated using the formula (length) × (width)$^2$/2, where the long axis diameter was regarded as the length and the short axis diameter was regarded as the

width. Mice were euthanized when the largest diameter of tumors reaches 20 mm or mice became moribund.

## Statistical analysis

Statistical analysis was performed using GraphPad Prism v6 and v8. Sample variance was tested using the F test. For samples with equal variance, the significance between the groups was analyzed by a two-tailed student's $t$-test. For samples with unequal variance, a two-tailed Welch's $t$-test was performed. For multiple comparisons, a one-way analysis of variance (ANOVA) with a post hoc Fisher's LSD test was used. The results were deemed significant at $0.01 <*P \le 0.05$, highly significant at $0.001 <**P \le 0.01$, and extremely significant at $***P \le 0.001$.

## Reporting summary

Further information on research design is available in the Nature Portfolio Reporting Summary linked to this article.

## Data availability

All data provided in this study can be found in the main text, figures, supplementary information, and source data files. Any additional requests for information can be directed to, and will be fulfilled by, the corresponding author. Source data are provided with this paper.

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

## Acknowledgements

H.W. acknowledges the financial support from NSF DMR 2143673 CAR, NIH R01CA274738, NIH R21CA270872, and the start-up package from the Department of Materials Science and Engineering at the University of Illinois at Urbana-Champaign and the Cancer Center at Illinois. X.S.Z. and H.W. acknowledge the financial support from the U.S. Defense Advanced Research Projects Agency (DARPA) Young Faculty Award (N660012314013). This material is based upon Q.C.'s work supported by the Air Force Office of Scientific Research under award number FA9550-23-1-0609. E.R.N. acknowledges support from NIH R01 CA234025. R.B. acknowledges the support from the National Institute of Biomedical Imaging and Bioengineering of the National Institutes of Health under Award Number T32EB019944. J.H. acknowledges the support from the Cancer Scholars for Translational and Applied Research (C*STAR) Pro-gram sponsored by the Cancer Center at Illinois and the Carle Cancer Center under Award Number CST EP012023.

## Author contributions

R.B. and H.W. conceived the study, designed the experiments and wrote the manuscript. J.H., Y.L., Y.B., D.L., J.Z., Y.W., E.R.N., Q.C., X.S.Z. and W.H. carried out the experiments and contributed to the revision of the manuscript.

## Competing interests

H.W., R.B., and J.H. filed a patent application for the metabolic tagging and targeting of extracellular vesicles (U.S. Provisional Patent Application No. 63/333,001). The remaining authors declare no competing interests.
