## [Peer Review File · Nature Communications]

Metabolic Tagging of Extracellular Vesicles and Development of Enhanced Extracellular Vesicle Based Cancer VaccinesREVIEWER COMMENTS

Reviewer #1 (Remarks to the Author):

Bhatta et al. described the method of metabolic tagging and targeted modulation of EVs and the use of this EV adjuvant to enhance the efficacy of tumor vaccines. The concept is interesting, and the effect in animal studies is promising. However, there are several quality control issues in the experiments. In addition, there is inappropriate information in the introduction as follows:

>Extracellular vesicles (EVs) such as exosomes inherit various cellular contents¹⁻³ and play a critical role in intercellular communication.

Although “exosomes” are a type of Extracellular vesicles (EVs), and they’re distinct families defined by biogenesis, there is currently no method distinctively separate microvesicles/ectosomes from exosomes due to the overlap and heterogeneity in sizes, thus it is strongly suggested that authors use the term “small EVs” instead of “exosomes” throughout the article. (reference:

<https://www.tandfonline.com/doi/full/10.1080/20013078.2018.1535750>)

>Genetic expression methods can introduce up to 105 proteins to the parent cell, but only a handful of them are inherited by the cell-secreted exosomes (10,000-fold surface area difference between a cell and an exosome), let alone the complexity and varied efficiency of genetic transfection.⁴⁻⁶

References 4-6 do not include such evidence or statement.

The genetic expression system will introduce a genetic expression cassette (DNA or RNA) but not 105 proteins to one parent cell (?). No such evidence is presented in the cited articles.

>In principle, exosomes may also bear lysine-containing proteins that are inherited from the parent cell, which again is minimal.

References 7-9 do not include such information. Also, it is unclear the relevance of lysine-containing proteins in this context or the exosomes. Two of the references do not mention lysine at all.

>For example, tumor exosomes can be endocytosed by antigen-presenting cells (e.g., DCs), followed by the processing and presentation of the encased antigens by DCs for subsequent priming of antigen-specific T cells.¹⁸⁻²⁰

References 18-20 do not mention endocytosis of tumor exosomes by antigen-presenting cells.

>Regarded as a safe source of tumor antigens, tumor exosomes exhibit desired pharmacokinetics than conventionally used dead tumor cells or tumor lysates, and have demonstrated the ability to induce antitumor cytotoxic T lymphocyte (CTL) response in multiple clinical trials.²¹⁻²³;

Reference 21 is a study using cell lines, which does not match the description. In addition, it is unclear and nonsense to compare the pharmacokinetics of exosomes to that of dead tumor cells or tumor lysate, which has no relevance to the clinical applications.

>As exosomes are endocytosed by DCs via endosomes where TLR3, TLR7, TLR8, and TLR9 are present,^{32,33}

Reference 32 is the study on the crystal structure of TLR3 ectodomain, and reference 33 describes Species-specific recognition of single-stranded RNA via toll-like receptors 7 and 8. These have no relevance to exosome endocytosis by DC.

Figure 2a-d: The authors described the images of control cells without Ac4ManAz treatment incubated in the text, but the figures are missing.

Figure S2: Please add units to each number and clarify the following:

Concentration (ug/L) of exosomes?

Conjugated Cy5 FI based on which amount?

Cy5 amount – of total or mole per liter?

Exosomes – per milliliter or liter?

Figure 3j and 3k: As mentioned previously, references 4-6 do not describe or provide evidence of the number of proteins per EV. In addition, the authors must report the method to determine the number of amines conjugated to an EV. Considering the surface area of

one particle, thousands of Ac4ManAz on a single EV are unrealistic. The assumption was based on one DBCO-Cy5 binds to a single Ac4ManAz, but multiple DBCO-Cy5 may bind to a single Ac4ManAz amplifying the signal. The authors should evaluate these carefully and provide data according to the claim.

Figure 4a and f: one of the concerns of the methods used in this work is the use of PBS buffers and DTT treatment known to cause massive aggregation of EVs.

(<https://doi.org/10.1038/s41598-019-47598-3>) The authors need to show the quality, especially sizes and concentrations of EVs post-processing, in addition to the fluorescence intensity.

Figure 4j: the resolution is too low, and the authors should provide larger, high-quality images. The fluorescence signal from Cy5 is way too high and large to claim signals from exosomes. The authors must provide evidence of EVs by staining them with EV markers.

Figure 4k: The figure shows the localization of Ac4ManAz and DBCO-Cy5 but the colocalization of EVs. Please provide evidence of the fluorescence signals derived from EV-associated Ac4ManAz, or add a control experiment by injecting Ac4ManAz alone and DBCO-Cy5 signals to show these signals derived from EV-associated Ac4ManAz.

Figure 6b: The blot for EV markers must be shown alongside the OVA protein, using the marker for cell-specific proteins, indicating no cell-debris contamination.

The manuscript becomes acceptable upon addressing the above comments.

Reviewer #2 (Remarks to the Author):

General comments:

Bhatta et al. describe an approach to efficaciously use tumor exosomes as a cancer vaccine. The authors' strategy specifically uses metabolic glycoengineering and downstream click chemistry of adjuvant to potentiate the exosome approach, increase immunostimulation, and drive vaccine efficacy. Notably, the authors demonstrate a rather surprising and

compelling result of converting a sub-stimulatory approach to a stimulatory one using their approach. However, it is unclear how or why sub-stimulatory conversion occurs (e.g., subcellular localization differences and trafficking vs. insufficient intracellular uptake in APCs without chemical conjugation of the adjuvant, etc.). Adding more mechanistic insights or discussion on how exosome conjugation of adjuvant alters immune cell interactions in such a way that would confer this substantial increase in antigen presentation would improve the manuscript substantially. Nevertheless, I find that the novelty of the approach and impressive efficacy results are sufficient quality and impact of Nature Communications and constitute a valuable contribution to the cancer vaccine development field. Additionally, the main findings and methods described are justified by the data and appropriate. Still, I recommend the authors address the general comments above and the specific changes/topics addressed below before publication.

Specific Comments:

1. While the metabolic labeling efficiency results of the cancer cells in Figure 2 are compelling, this base approach is already established. I suggest text and figure changes to the authors to minimize previously established results and focus on this strategy as a necessary element of exosome (such as the tagging characterization and functional results of generating exosomes from each line with the chemical tag and their downstream enhancement of immunostimulation). Perhaps combine figures 1 and 2 and move the base labeling efficiency to the supplement.
2. The text refers to the size and morphology of exosomes for figures 2 and 3 and that no differences were observed with and without the metabolic labeling. The authors describe this several times and state that this is an important result. Why is this significant? The authors should discuss this result in more detail as to why this is a meaningful/ substantial evaluation method of cancer exosome potency for cancer vaccine development.
3. What is the purpose of metabolic glycoengineering of MSCs, DCs, and T cells? The authors just seem to show they can accomplish this but do not show why this is useful. In general, it is unclear what Figure 3 contributes to this manuscript. If the authors wish to include this, I would suggest adding in a functional study of where this can be applied (e.g., demonstration of immunomodulatory effects of tagged exosomes from MSCs on an immune cell). As it is, it comes across as an aside that seems out of place for a paper ostensibly about using metabolic glycoengineering to enhance exosome-based cancer vaccines. If the authors

cannot address this, I suggest simply removing this from the manuscript (it is already 7 figures)/moving to the supplement.

4. Exosomes were already isolated via ultracentrifugation in the manuscript. Why is the bead capture approach necessary? It is unclear if this is utilized anywhere else in the manuscript or why it is included. The authors need to provide a rationale or comparison of yield/quality etc. for why this approach was developed.

5. In general, figures 5-6 and part of 7 are quite compelling and convincing of the significant advancement of this work.

6. In general, the authors show impressive increased activation of DCs through conjugation of the CPG adjuvant. Could the authors expand further upon the mechanism for this?

7. What is the author's interpretation of why Exo-N3 is more activating than Exo-N3+CPG in fig6j-l? This result seems surprising and warrants discussion in the manuscript. Additionally, the level of OVA-specific T cell activation differences appears quite small, given the robust differences in survival. Can the authors discuss this and how these results compare to previous cancer vaccine literature?

8. Figure 7 is somewhat poorly labeled and organized. The figure would be improved by labeling directly in the figure what are the OVA vs. B16F10 melanoma results, providing schematics for both efficacy studies, and clearly grouping the different studies together. Also, the toxicity data seems unnecessary and also a bit odd to present "no toxicity" differences in a cancer vaccine efficacy study. I would suggest the authors move all to the supplement.

Technical comments:

1. Figure 3j-k is unclear what the blue and colored circles represent in terms of data. What are these values? This needs significant clarification/improvement for Nat. Comm. standards.

2. Kaplan-Meier plots throughout are labeled "probability of survival." How is that different from survival? This comes across as confusing. If this is not actual survival, the authors should better explain this discrepancy. If it is actual survival, as it should be, the authors should relabel it as "Percent Survival," as is standard for these plots.

3. Figure legends should include N clarification (number of animals vs. technical replicates for assays, etc.)

4. Alternations of SEM and SD throughout are not explained or justified. 95% confidence

intervals would be preferred for true biological replicates as it is the least ambiguous.

Reviewer #3 (Remarks to the Author):

This manuscript presents a metabolic tagging technology to install chemical tags on extracellular vesicles with a higher magnitude than conventional methods. This technology allows the tracking of the vesicles and more importantly to modulate the immunogenicity of the vesicles by adding the TLR9 agonist CpG to activate dendritic cells. Using these modified exosomes as an anti-cancer vaccine, the authors showed an impressive tumor control both in prophylactic and therapeutic settings.

Despite these interesting aspects, I see some important drawbacks.

Major comments:

1) The tumor growth is completely controlled by the Exo-CpG treatment. However, the percentage of specific T cells is very low (around 2% tetramer positive cells), and even if it is significantly increased compared to other groups, the difference remains low, so it seems surprising that these few cells are solely responsible for this tumor control.

This specific T cells measurement was only performed in the blood of the mice. It will be important to look at the presence of these specific T cells also in tumors, or in lymphoid organs, and see if a bigger difference can be observed there. The phenotype analysis of these T cells should also be performed, for example to evaluate if they have a stronger activation status, an increased memory phenotype, or if they express less exhaustion markers, compared to T cells induced by the other treatments.

2) An important control is lacking from this study. The authors showed that their metabolic approach is much more efficient compared to conventional genetic approaches to express tags on the exosomes. But, they should then also show functionally, in vitro or in vivo, the advantages of their method, compared to conventional approaches of tagging.

3) The OVA-expressing tumor models are highly immunogenic models as they express an exogenous protein. The authors showed nice results using the B16-F10 melanoma model. It would be interesting to also look if in this less immunogenic model if the presence of specific T cells such as CD8 T cells specific for the trp2 or gp100 antigens is also increase by

Exo-CpG.

4) The authors demonstrated the efficacy of their Exo-CpG compared to aPD-1 alone or aPD1 + naked exosome, but they didn't try to combine their Exo-CpG to the aPD-1. It would be interesting to demonstrate the synergy of the Exo-CpG with the aPD1 treatment, for potential future combinations with immunotherapies.

Minor comments:

1) Can the authors also show the control confocal images of the cancer cell lines treated without Ac4ManAZ (Fig2a-d).

2) How is calculated the number of Tags per exosome? (Fig 3J). Also, the number of tags is similar, but the fluorescence intensity of Cy5 on the MSC or on exosomes derived from MSC is higher than on the dendritic cells or T cells (Fig3). Is there a differential level of expression or intensity of the signal depending on the cell type from which the exosomes are derived?

3) The figure S4a is not mentioned in the text.

4) The methods section can be a little more detailed. For example:

- please indicate the reference of the protocol you are mentioning for the BMDC differentiation.

- how is calculated the proliferation index of the CFSE T cell in the in vitro proliferation assay?

- The sequence of the OVA peptide is SIINFEKL and not SINFEKL (typo in materials and instrumentation section: "SINFEKL-MHCI tetramer was requested from the NIH Tetramer").

Point-by-point reply to reviewers' comments

(All responses were colored in blue; all changes in the manuscript were highlighted in yellow)

Reviewer 1

Bhatta et al. described the method of metabolic tagging and targeted modulation of EVs and the use of this EV adjuvant to enhance the efficacy of tumor vaccines. The concept is interesting, and the effect in animal studies is promising. However, there are several quality control issues in the experiments. In addition, there is inappropriate information in the introduction as follows:

Response: We thank the Reviewer for the positive comments and have addressed the questions accordingly below.

(1) Extracellular vesicles (EVs) such as exosomes inherit various cellular contents¹⁻³ and play a critical role in intercellular communication. Although “exosomes” are a type of Extracellular vesicles (EVs), and they’re distinct families defined by biogenesis, there is currently no method distinctively separate microvesicles/ exosomes from exosomes due to the overlap and heterogeneity in sizes, thus it is strongly suggested that authors use the term “small EVs” instead of “exosomes” throughout the article.

(reference: <https://www.tandfonline.com/doi/full/10.1080/20013078.2018.1535750>)

Response: We have now replaced ‘exosome’ with ‘EV’ throughout the revised manuscript.

(2) Genetic expression methods can introduce up to 10^6 proteins to the parent cell, but only a handful of them are inherited by the cell-secreted exosomes (10,000-fold surface area difference between a cell and an exosome), let alone the complexity and varied efficiency of genetic transfection.⁴⁻⁶ References 4-6 do not include such evidence or statement. The genetic expression system will introduce a genetic expression cassette (DNA or RNA) but not 10^6 proteins to one parent cell (?). No such evidence is presented in the cited articles.

Response: We thank the Reviewer for this comment. We intended to state that “as the density of cell-surface proteins is at the scale of 10^3 - 10^6 per cell, genetic methods that utilize the cellular protein expression machinery can introduce up to 10^6 proteins per cell”.

The reported number density of cell-surface proteins varies from paper to paper. Using EGFR (a well-studied cell-surface protein) as an example, we summarize some literature reports below:

- a) “the total number of EGF receptors per cell of different cell lines varied from 1.6×10^3 sites/cell (for A1Ab 496) to 1.5×10^6 sites/cell (for BT-20)” (Cancer Res, 1982, 42; 4394).
- b) “the binding and internalization of epidermal growth factor (EGF) in human epithelioid carcinoma cells (A-431), which have approximately 2.6×10^6 receptors per cell, has been followed with 125 I-labeled EGF and by fluorescence microscopy” (PNAS, 1978, 75 (7), 3317-3321).
- c) “it was calculated from these data that each human fibroblast cell was capable of binding approximately 97,000 molecules of monoiodo-EGF” (The Journal of Biological Chemistry, 1975, 250 (11), 4297-4304).
- d) “These results are consistent with observed EGFR copy numbers of 1.2×10^6 and 8.7×10^4 for A431 and SK-BR-3 cells” (Cancer Research 1990, 50, 1550-1558).

- e) “the surface of a typical cell bears 10,000 – 20,000 receptors for a particular hormone” (<http://www.ncbi.nlm.nih.gov/books/NBK21553/>).

Similar ranges (10^3 - 10^6) were also reported for other surface proteins. To avoid confusion, we have now updated the text: “As the density of proteins expressed on the cell membrane is often in the range of 10^3 - 10^6 per cell,^{4,7} genetic expression methods can potentially introduce up to 10^6 proteins to the parent cell, but only a small quantity are inherited by the cell-secreted EVs (10,000-fold surface area difference between a cell and an EV), let alone the complexity and varied efficiency of genetic transfection” in the revised manuscript.

(3) In principle, exosomes may also bear lysine-containing proteins that are inherited from the parent cell, which again is minimal. References 7-9 do not include such information. Also, it is unclear the relevance of lysine-containing proteins in this context or the exosomes. Two of the references do not mention lysine at all.

Response: We thank the Reviewer for this comment. We meant to describe that previous research has utilized the surface amine groups of EVs for chemical modification (and lysine is the type of amino acid that bears amine groups), and the cited references all used amine-carboxyl chemistry for the functionalization of EVs. To avoid confusion, we have now revised the text into “In principle, EVs may also bear amine-bearing proteins that are inherited from the parent cell for surface functionalization using amine-carboxyl chemistry,¹¹⁻¹⁴ but the number of surface amine groups per EV could be minimal.” We have also added the references (new Ref. 11-14) describing the use of amine-carboxyl chemistry to functionalize or modify EVs.

11 Smyth T., Petrova K., Payton N.M., Persaud I., Redzic J.S., Graner M.W., Smith-Jones P., Anchordoquy T.J. Surface Functionalization of Exosomes Using Click Chemistry. *Bioconjugate Chem* **25**, 1777–1784 (2014)

12 Si, Y.; Kim, S.; Zhang, E.; Tang, Y.; Jaskula-Sztul, R.; Markert, J.M.; Chen, H.; Zhou, L.; Liu, X. Targeted Exosomes for Drug Delivery: Biomanufacturing, Surface Tagging, and Validation. *Biotechnol J* **15**, 1900163 (2020)

13 Hosseini, N.F.; Amini, R.; Ramezani, M.; Saidijam, M.; Hashemi, S.M.; Najafi, R. AS1411 aptamer-functionalized exosomes in the targeted delivery of doxorubicin in fighting colorectal cancer. *Biomed Pharm* **155**, 113690 (2022)

14 Johnson, V.; Vasu, S.; Kumar, U.S.; Kumar, M. Surface-Engineered Extracellular Vesicles in Cancer Immunotherapy. *Cancers* **15**, 2838 (2023)

(4) For example, tumor exosomes can be endocytosed by antigen-presenting cells (e.g., DCs), followed by the processing and presentation of the encased antigens by DCs for subsequent priming of antigen-specific T cells.¹⁸⁻²⁰ References 18-20 do not mention endocytosis of tumor exosomes by antigen-presenting cells.

Response: The original Ref 18-20 talked about the uptake of EVs by dendritic cells and the subsequent processing and presentation of EV-encased antigens by dendritic cells, which is an essential step for EVs to function as a vaccine. However, we understand that the Reviewer might be asking for references that directly show the EV endocytosis data, so we have now included new references (Ref. 23-26) that directly show the confocal images of EV uptake by dendritic cells.

23. Montecalvo A, Shufesky W, Stolz DB, Sullivan M, Wang Z, Divito S, Papworth G, Watkins S, Robbins P, Larregina A, Morelli A. Exosomes As a Short-Range Mechanism to Spread Alloantigen between Dendritic Cells during T Cell Allorecognition. *Journal of immunology*. 2008, 180 (5), 3081-90. [Talks about internalization of exosomes by DC and have confocal images to verify it]

24. Morelli, A, Larregina A, Shufesky W, Sullivan M, Stolz DB, Papworth G, Zahorchak A, Logar A, Wang Z, Watkins S, Falo L, Thomson A. Endocytosis, intracellular sorting, and processing of exosomes by dendritic cells. *Blood* 2004, 104 (10), 3257-3266. [Demonstrates the effective internalization of exosomes by DC (confocal imaging) and further processing and presentation of peptide antigens]

25. Torralba, D., Baixauli, F., Villarroya-Beltri, C. *et al.* Priming of dendritic cells by DNA-containing extracellular vesicles from activated T cells through antigen-driven contacts. *Nat Commun* **9**, 2658 (2018). [Contains confocal images of EV uptake by dendritic cells]

26. Morales-Kastresana, A., Telford, B., Musich, T.A. *et al.* Labeling Extracellular Vesicles for Nanoscale Flow Cytometry. *Sci Rep* **7**, 1878 (2017). [Contains confocal images of EV uptake by dendritic cells]

(5) Regarded as a safe source of tumor antigens, tumor exosomes exhibit desired pharmacokinetics than conventionally used dead tumor cells or tumor lysates, and have demonstrated the ability to induce antitumor cytotoxic T lymphocyte (CTL) response in multiple clinical trials.²¹⁻²³; Reference 21 is a study using cell lines, which does not match the description. In addition, it is unclear and nonsense to compare the pharmacokinetics of exosomes to that of dead tumor cells or tumor lysate, which has no relevance to the clinical applications.

Response: We appreciate this comment from the Reviewer. Our intention was to highlight the well-documented safety profile of EVs. To avoid confusion, we have now revised the text into “Regarded as a safer source of tumor antigens than conventionally used dead tumor cells or tumor lysates, tumor EVs have demonstrated the ability to induce antitumor cytotoxic T lymphocyte (CTL) response in clinical trials”. The original ref 21 has also been removed.

(6) As exosomes are endocytosed by DCs via endosomes where TLR3, TLR7, TLR8, and TLR9 are present,^{32,33} Reference 32 is the study on the crystal structure of TLR3 ectodomain, and reference 33 describes Species-specific recognition of single-stranded RNA via toll-like receptors 7 and 8. These have no relevance to exosome endocytosis by DC.

Response: The original 32 and 33 were indeed intended for introducing more information of TLRs. We now have also included the references (Ref. 23-26) that provide direct evidence of EV endocytosis by dendritic cells.

23. Montecalvo A, Shufesky W, Stolz DB, Sullivan M, Wang Z, Divito S, Papworth G, Watkins S, Robbins P, Larregina A, Morelli A. Exosomes As a Short-Range Mechanism to Spread Alloantigen between Dendritic Cells during T Cell Allorecognition. *Journal of immunology*. 2008, 180 (5), 3081-90. [Talks about internalization of exosomes by DC and have confocal images to verify it]

24. Morelli, A, Larregina A, Shufesky W, Sullivan M, Stolz DB, Papworth G, Zahorchak A, Logar A, Wang Z, Watkins S, Falo L, Thomson A. Endocytosis, intracellular sorting, and

processing of exosomes by dendritic cells. *Blood* 2004, 104 (10), 3257-3266. [Demonstrates the effective internalization of exosomes by DC (confocal imaging) and further processing and presentation of peptide antigens]

25. Torralba, D., Baixauli, F., Villarroya-Beltri, C. *et al.* Priming of dendritic cells by DNA-containing extracellular vesicles from activated T cells through antigen-driven contacts. *Nat Commun* 9, 2658 (2018). [Contains confocal images of EV uptake by dendritic cells]

26. Morales-Kastresana, A., Telford, B., Musich, T.A. *et al.* Labeling Extracellular Vesicles for Nanoscale Flow Cytometry. *Sci Rep* 7, 1878 (2017). [Contains confocal images of EV uptake by dendritic cells]

(7) Figure 2a-d: The authors described the images of control cells without Ac₄ManAz treatment incubated in the text, but the figures are missing.

Response: We thank the Reviewer for this comment. We have now added the control images of cells without Ac₄ManAz treatment.

Figure 2. (a-d) CLSM images of (a) 4T1 breast cancer cells, (b) LS174T colon cancer cells, (c) GL261 glioblastoma cells, and (d) BxPC-3 pancreatic cancer cells, respectively, after treated with Ac₄ManAz or PBS for three days and incubated with DBCO-Cy5 (red) for 30 min.

(8) Figure S2 (now Fig. S3): Please add units to each number and clarify the following: Concentration (ug/L) of exosomes? Conjugated Cy5 FI based on which amount? Cy5 amount – of total or mole per liter? Exosomes – per milliliter or liter?

Response: We have now added units to the numbers and clarified what each value means. In particular, the unit of fluorescence intensity (FI) is a.u. (arbitrary unit). The FI of DBCO-Cy5 conjugated to EVs was measured on a plate reader and subtracted with the background FI. Based on the standard curve of Cy5 (FI versus concentration), we calculated the number of Cy5 molecules being conjugated to EVs. As we can quantify the number of EVs in the solution, the number of Cy5 molecules per EV can thus be calculated. This allowed us to estimate the number of surface azido groups per EV (because not all azides can be consumed by DBCO-Cy5, the number of azides per EV is expected to be higher than the calculated Cy5 number density).

Fig. S3. Quantification of surface azido groups per EV. EVs were collected from Ac₄ManAz- or PBS-treated E.G7-OVA cells and incubated with DBCO-Cy5 for 30 min. A standard curve of Cy5 fluorescence intensity was used to calculate the amount of conjugated Cy5 molecules, as a means to estimate the number of azido groups per EV.

(9) Figure 3j and 3k: As mentioned previously, references 4-6 do not describe or provide evidence of the number of proteins per EV. In addition, the authors must report the method to determine the number of amines conjugated to an EV. Considering the surface area of one particle, thousands of Ac₄ManAz on a single EV are unrealistic. The assumption was based on one DBCO-Cy5 binds to a single Ac₄ManAz, but multiple DBCO-Cy5 may bind to a single Ac₄ManAz amplifying the signal. The authors should evaluate these carefully and provide data according to the claim.

Response: As described in our response to comment #8 above, we subtracted our measured Cy5 fluorescence intensity (FI) with the FI of non-specifically bounded DBCO-Cy5, and then utilized this net Cy5 FI to estimate the number of Cy5 molecules per EV. DBCO-Cy5 and azido group can react at 1:1 molar ratio via the efficient click chemistry. It is true that not all azido groups can be consumed by DBCO-Cy5, which is why we think the actual density of azido groups should be higher than the density estimated at the 100% consumption rate of azido groups. Considering the size of EVs (~100 nm), it is not uncommon to have a few thousand small-molecule tags like azido groups per EV.

Regarding the number density of amine groups on EVs, we conjugated Cy5-NHS to EVs (assuming the presence of amine groups on the surface), quantified Cy5 FI, calculated the number of Cy5 molecules being conjugated, and then estimated the number of amine groups per EV. After carefully considering all Reviewers' comments, we agree that there is a lack of a reliable method to quantify the number of amine groups and protein molecules on the surface of EVs. The comparison of the estimated numbers could cause a lot of confusion to the readers. Therefore, we have now deleted the original Fig. 3j-k.

(10) Figure 4a and f: one of the concerns of the methods used in this work is the use of PBS buffers and DTT treatment known to cause massive aggregation of EVs. (<https://doi.org/10.1038/s41598-019-47598-3>) The authors need to show the quality, especially sizes and concentrations of EVs post-processing, in addition to the fluorescence intensity.

Response: The paper the Reviewer mentioned actually talked about the aggregation of proteins instead of EVs as a result of PBS or DTT treatment (30-minute incubation at $>37^{\circ}\text{C}$), which the authors noticed during the western blot analysis of proteins. In our experiment, EVs were conjugated to magnetic beads via a reductive-cleavable disulfide (S-S), and then incubated with DTT that can cleave the disulfide bond for 10 minutes at room temperature to release the conjugated EVs. The purpose of this experiment was to demonstrate that one can retrieve (i.e., recover) nanosized EVs using our metabolic labeling method. However, to fully answer the Reviewer's question on whether DTT treatment could cause the aggregation of EVs. We have now provided the DLS data of EVs before and after DTT treatment (10 minutes at room temperature) (Fig. S5), which did not show any noticeable differences. We have added the following text in the revised manuscript: "It is noteworthy that DTT treatment did not disrupt or alter the size of EVs (Fig. S5a-c)." Fig. S5 has also been added to the supplementary figures.

Figure S5. Representative size distribution of 4T1-derived EVs treated with (a) 10 mM DTT or (b) PBS for 10 minutes at room temperature. (c) Average diameter of EVs treated with DTT or PBS for 10 minutes.

(11) Figure 4j: the resolution is too low, and the authors should provide larger, high-quality images. The fluorescence signal from Cy5 is way too high and large to claim signals from exosomes. The authors must provide evidence of EVs by staining them with EV markers.

Response: We have provided a new confocal image for the uptake of EVs by dendritic cells (new Fig. 3j). We have also stained EVs with FITC-conjugated anti-tsg101, a commonly used EV marker, and visualized the uptake of anti-tsg101-stained EVs by dendritic cells under a confocal microscope (new Fig. 3k).

We have now added the new Fig. 3j-k and updated the following text in the revised manuscript: “Confocal imaging confirmed the internalization of Cy5-conjugated EVs by DCs (Fig. 3j). Successful staining of EVs with FITC-conjugated anti-tsg101, a commonly used EV marker, further confirmed the uptake of EVs by DCs (Fig. 3k).”

Figure (3j) CLSM images of BMDCs after 60-min incubation with Cy5-conjugated EVs. Cell nuclei and membranes were stained with DAPI (blue) and Cell Mask Stain (green), respectively. Scale bar: 10 μ m. (3k) CLSM images of BMDCs after 60-min incubation with anti-tsg101-conjugated EVs. Cell nuclei were stained with DAPI (blue). FITC-conjugated anti-tsg101 was used to stain EVs. Scale bar: 10 μ m.

(12) Figure 4k (now Fig. 3l): The figure shows the localization of Ac4ManAz and DBCO-Cy5 but the colocalization of EVs. Please provide evidence of the fluorescence signals derived from EV-associated Ac4ManAz, or add a control experiment by injecting Ac4ManAz alone and DBCO-Cy5 signals to show these signals derived from EV-associated Ac4ManAz.

Response: We would like to clarify a bit more on what Fig. 3l (original Fig. 4k) means, which might have created the confusion. For Fig. 3l, we actually injected Cy5-conjugated EVs (synthesized via the conjugation of DBCO-Cy5 to azido-tagged EVs *in vitro*) or non-conjugated EVs subcutaneously into the flank of mice, and then tried to confirm the migration of injected EVs

into the lymph nodes. The Cy5 signal in the lymph nodes confirmed the successful migration of injected Cy5-conjugated EVs to the lymph nodes, which is an important step for subsequent immune modulation.

(13) Figure 6b (now Fig. 5b): The blot for EV markers must be shown alongside the OVA protein, using the marker for cell-specific proteins, indicating no cell-debris contamination.

Response: We thank the Reviewer for this comment. We have now performed the western blot analysis of CD63 and TSG 101 proteins, two commonly used EV markers, for E.G7-OVA derived EVs. The western blot image has been updated in Fig. 5b. We have added the following text in the revised manuscript: “We also successfully detected the presence of tsg101 and CD63 proteins, two commonly used EV markers, in E.G7-OVA derived EVs (Fig. 5b).”

Figure 5b. Western Blot analysis of OVA protein from E.G7-OVA EVs, B16F10 EVs, or pure OVA control. Also shown are the analysis of Tsg101 and CD63 proteins, two commonly used EV markers, from E.G7-OVA EVs.

Reviewer #2

General comments:

Bhatta et al. describe an approach to efficaciously use tumor exosomes as a cancer vaccine. The authors' strategy specifically uses metabolic glycoengineering and downstream click chemistry of adjuvant to potentiate the exosome approach, increase immunostimulation, and drive vaccine efficacy. Notably, the authors demonstrate a rather surprising and compelling result of converting a sub-stimulatory approach to a stimulatory one using their approach. However, it is unclear how or why sub-stimulatory conversion occurs (e.g., subcellular localization differences and trafficking vs. insufficient intracellular uptake in APCs without chemical conjugation of the adjuvant, etc.). Adding more mechanistic insights or discussion on how exosome conjugation of adjuvant alters immune cell interactions in such a way that would confer this substantial increase in antigen

presentation would improve the manuscript substantially. Nevertheless, I find that the novelty of the approach and impressive efficacy results are sufficient quality and impact of Nature Communications and constitute a valuable contribution to the cancer vaccine development field. Additionally, the main findings and methods described are justified by the data and appropriate. Still, I recommend the authors address the general comments above and the specific changes/topics addressed below before publication.

Response: We thank the Reviewer for the positive comments, and have addressed the questions accordingly below.

(1) While the metabolic labeling efficiency results of the cancer cells in Figure 2 are compelling, this base approach is already established. I suggest text and figure changes to the authors to minimize previously established results and focus on this strategy as a necessary element of exosome (such as the tagging characterization and functional results of generating exosomes from each line with the chemical tag and their downstream enhancement of immunostimulation). Perhaps combine figures 1 and 2 and move the base labeling efficiency to the supplement.

Response: We appreciate this comment from the Reviewer. As we intend to describe an overview of the project in Figure 1, we feel it might be easier for the readers to follow if we keep Figure 1 and Figure 2 separate. We hope to get the understanding of the Reviewer, but are open to combine Figures 1-2 if the Reviewer strongly suggests that way.

(3) The text refers to the size and morphology of exosomes for figures 2 and 3 and that no differences were observed with and without the metabolic labeling. The authors describe this several times and state that this is an important result. Why is this significant? The authors should discuss this result in more detail as to why this is a meaningful/ substantial evaluation method of cancer exosome potency for cancer vaccine development.

Response: We thank the Reviewer for this comment. The phenomenon that exosomes maintain their size and morphology regardless of metabolic labeling indicates that the labeling process itself does not alter the biogenesis of exosomes. This could be critical in a sense that our approach does not impair the intrinsic properties of exosomes, and that the observed benefits in cancer vaccination (Figs. 4-6) are not a result of the alteration on the structure of exosome itself.

(3) What is the purpose of metabolic glycoengineering of MSCs, DCs, and T cells? The authors just seem to show they can accomplish this but do not show why this is useful. In general, it is unclear what Figure 3 contributes to this manuscript. If the authors wish to include this, I would suggest adding in a functional study of where this can be applied (e.g., demonstration of immunomodulatory effects of tagged exosomes from MSCs on an immune cell). As it is, it comes across as an aside that seems out of place for a paper ostensibly about using metabolic glycoengineering to enhance exosome-based cancer vaccines. If the authors cannot address this, I suggest simply removing this from the manuscript (it is already 7 figures)/moving to the supplement.

Response: The initial purpose of Fig. 3 was to show that our exosome tagging technology can be universally applied to different types of cells. However, we agree with the Reviewer that this

Figure might not be critical for this manuscript, and have now moved it to the supplement as Fig. S2.

(4) Exosomes were already isolated via ultracentrifugation in the manuscript. Why is the bead capture approach necessary? It is unclear if this is utilized anywhere else in the manuscript or why it is included. The authors need to provide a rationale or comparison of yield/quality etc. for why this approach was developed.

Response: Sorry for the confusion here. Fig. 3a-f (original Fig. 4a-f) are to demonstrate that the metabolically tagged exosomes can be conjugated onto magnetic beads and pulled down by a magnet, and can further be recycled via the incorporation of a cleavable bond (e.g., disulfide bond). The purpose of these experiments/data are to further validate the chemical tagging of exosomes and demonstrate that the surface chemical tags of exosomes could be useful for isolation and purification of exosomes. We agree with the Reviewer that this piece of data are not relevant to the vaccination data in Fig. 4-6, but they could be of interest to researchers working on exosome-based diagnostics.

(5) In general, figures 5-6 and part of 7 are quite compelling and convincing of the significant advancement of this work.

Response: We thank the Reviewer for this positive comment.

(6) In general, the authors show impressive increased activation of DCs through conjugation of the CpG adjuvant. Could the authors expand further upon the mechanism for this?

Response: One of our hypotheses is that our metabolic tagging approach can install a high density of TLR9 agonist (CpG) on the surface of exosomes. Once the CpG-conjugated exosomes are endocytosed by DCs via endosomes, the sufficient CpG on exosomes (i.e., a nanocluster of CpG molecules) can effectively stimulate the TLR9 scattered on the surface of endosomes, resulting in the superior activation of DCs. We have now added the following text to the Discussion section: “One of our hypotheses is that the metabolic tagging approach can install a high density of TLR9 agonist (CpG) on the surface of EVs. Once the CpG-conjugated EVs are endocytosed by DCs via endosomes, the nanocluster of CpG on the surface of EVs can effectively stimulate the TLR9 scattered on the surface of endosomes, resulting in the timely activation of DCs.”

(7) What is the author’s interpretation of why Exo-N3 is more activating than Exo-N3+CPG in fig6j-l? This result seems surprising and warrants discussion in the manuscript. Additionally, the level of OVA-specific T cell activation differences appears quite small, given the robust differences in survival. Can the authors discuss this and how these results compare to previous cancer vaccine literature?

Response: We thank the Reviewer for this careful catch. In Fig 5j-l (original Fig. 6j-l), the difference between Exo-N₃ and Exo-N₃ + CpG is actually statistically non-significant ($p > 0.05$). This is consistent at different time points, demonstrating the negligible differences between the two groups. Regarding the question on the level of OVA-specific T cells, it is not uncommon to only see a few percent of tetramer⁺ or IFN- γ ⁺ CD8⁺ T cells in PBMCs while obtaining significant

antitumor efficacy data (Ref. 41, 44, 51, 52), which is partially related to the sensitivity of the detection method for antigen-specific CD8⁺ T cells. Often, we need to combine the tumor data and CD8⁺ T cell data to fully assess the effectiveness of a cancer vaccine.

(8) Figure 7 is somewhat poorly labeled and organized. The figure would be improved by labeling directly in the figure what are the OVA vs. B16F10 melanoma results, providing schematics for both efficacy studies, and clearly grouping the different studies together. Also, the toxicity data seems unnecessary and also a bit odd to present “no toxicity” differences in a cancer vaccine efficacy study. I would suggest the authors move all to the supplement.

Response: We thank the Reviewer for the great suggestion. We have now re-organized Figure 6 (original Figure 7) by directly labeling tumor type on the figures, adding a timeframe for B16F10 tumor study, and moving the H&E data to the supplement.

Figure 6

(9) Technical comments:

1. Figure 3j-k is unclear what the blue and colored circles represent in terms of data. What are these values? This needs significant clarification/improvement for Nat. Comm. standards.

Response: The initial Fig 3j showed the comparison of average number of surface tags per exosome obtained using conventional methods or our metabolic tagging approach. The initial Fig 3k shows the comparison of number of protein molecules that can be introduced via the conventional gene expression method or chemical tags that can be introduced by our metabolic labeling method. In alignment with other Reviewers' comments, we have now deleted the original Fig. 3j-k, as there is no reliable method to obtain an accurate estimate of the number density of proteins or amine groups on the surface of exosomes.

2. Kaplan-Meier plots throughout are labeled "probability of survival." How is that different from survival? This comes across as confusing. If this is not actual survival, the authors should better explain this discrepancy. If it is actual survival, as it should be, the authors should relabel it as "Percent Survival," as is standard for these plots.

Response: We indeed meant the overall survival. We have now changed the label to 'Percent Survival' for all Kaplan-Meier plots.

3. Figure legends should include N clarification (number of animals vs. technical replicates for assays, etc.)

Response: We have clarified the number of animals and technical replicates in the figure captions.

4. Alternations of SEM and SD throughout are not explained or justified. 95% confidence intervals would be preferred for true biological replicates as it is the least ambiguous.

Response: Thanks for this comment. We used mean \pm SD for all the numerical data except for the tumor volume data where we used mean \pm SEM, which is commonly used in other cancer immunotherapy papers. We stated the use of SD or SEM for each Figure, provided the statistical analysis method for each figure, and also provided p values for all comparisons.

Reviewer #3

This manuscript presents a metabolic tagging technology to install chemical tags on extracellular vesicles with a higher magnitude than conventional methods. This technology allows the tracking of the vesicles and more importantly to modulate the immunogenicity of the vesicles by adding the TLR9 agonist CpG to activate dendritic cells. Using these modified exosomes as an anti-cancer vaccine, the authors showed an impressive tumor control both in prophylactic and therapeutic settings. Despite these interesting aspects, I see some important drawbacks.

Response: We thank the reviewer for the positive comments, and have addressed the questions accordingly below.

Major comments:

1) The tumor growth is completely controlled by the Exo-CpG treatment. However, the percentage of specific T cells is very low (around 2% tetramer positive cells), and even if it is significantly increased compared to other groups, the difference remains low, so it seems surprising that these few cells are solely responsible for this tumor control. This specific T cells measurement was only performed in the blood of the mice. It will be important to look at the presence of these specific T cells also in tumors, or in lymphoid organs, and see if a bigger difference can be observed there. The phenotype analysis of these T cells should also be performed, for example, to evaluate if they have a stronger activation status, an increased memory phenotype, or if they express less exhaustion markers, compared to T cells induced by the other treatments.

Response: We thank the Reviewer for this comment. Regarding the question on the level of OVA-specific T cell activation, it is not uncommon to only see a few percent of tetramer⁺ or IFN- γ ⁺ CD8⁺ T cells in PBMCs while obtaining significant antitumor efficacy data. Often, we need to combine the tumor data and CD8⁺ T cell data to fully assess the effectiveness of a cancer vaccine. While analyzing the antigen-specific CD8⁺ T cells in the blood at varied times is a standard way to monitor the persistence of T cell responses, per the suggestion of the Reviewer, we have supplemented a study to analyze the numbers, activation status, and phenotypes of T cells and other immune cells in the tumors and tumor-draining lymph nodes.

We have now added the new data as Fig. S16-17, and added the following text in the revised manuscript: “To better understand the alteration of the tumor microenvironment as a result of EV vaccination, we also analyzed the number, activation status, and memory phenotype of T cells in the tumors and tumor-draining lymph nodes at 6 days post vaccination (**Fig. S16a**). Compared to EV alone or the mixture of EVs and CpG, CpG-conjugated EVs resulted in a higher number of CD8⁺ T cells (**Fig. S16b-c**), a lower number of regulatory T (Treg) cells (**Fig. S16d**), and a higher CD8⁺/Treg ratio (**Fig. S16e**). The intratumoral CD8⁺ T cells also exhibited an upregulated expression of CD69 activation marker (**Fig. S16f**) and increased fraction of central memory phenotype (CD44⁺CD62L⁺) and effector memory phenotype (CD44⁺CD62L⁻) in mice treated with CpG-conjugated EVs, compared to mice treated with EV alone or the mixture of EVs and CpG (**Fig. S16g-h**). It is noteworthy that CpG-conjugated EVs also resulted in an increased expression of PD-1 by intratumoral CD8⁺ T cells than the control groups (**Fig. S16i**), which is consistent with previous reports that activated CD8⁺ T cells tend to express a higher level of exhaustion markers.^{44,50-52} In the tumor-draining lymph nodes, mice treated with CpG-conjugated EVs also showed a higher CD8/Treg ratio and an increased number of CD69⁺ CD8⁺ T cells and memory-phenotype CD8⁺ T cells, in comparison with mice treated with EV alone or the mixture of EV and CpG (**Fig. S17a-e**).”

Figure S16. EV vaccine alters the activation status and phenotype of CD8⁺ T cells in the tumor microenvironment. (a) Timeframe of *in vivo* study. B16F10 tumor was subcutaneously inoculated on day 0. CpG-conjugated EVs, the mixture of CpG and EVs, or EV alone (n= 5 per group) were subcutaneously injected on day 11. Tumors were isolated for immune cell analysis on day 17. (b) Percentages of CD45⁺CD3⁺ T cells in tumors. (c) Percentages of CD8⁺ T cell in tumors. (d) Percentages of FoxP3⁺ cells among CD4⁺ T cells in tumors. (e) CD8⁺ T/Treg number ratios in tumors. (f) Percentages of CD69⁺ cells among CD8⁺ T cells in tumors. Also shown are the percentages of (g) central memory T cells (CD44⁺ CD62L⁺) and (h) effector memory T cells (CD44⁺ CD62L⁻) in tumors. (i) Percentages of PD-1⁺ cells among CD8⁺ T cells in tumors. All the numerical data are presented as mean \pm SD (0.01 < *P* \leq 0.05; ***P* \leq 0.01; ****P* \leq 0.001; *****P* \leq 0.0001).

Figure S17. EV vaccine alters the activation status and phenotype of CD8⁺ T cells in the tumor-draining lymph nodes. CpG-conjugated EVs, the mixture of CpG and EVs, or EV alone (n= 5) were subcutaneously injected on day 11. Tumor-draining lymph nodes were isolated for immune cell analysis on day 17. (a) CD8⁺ T/Treg number ratios in lymph nodes. (b) Percentages of CD69⁺ cells among CD8⁺ T cells in lymph nodes. Also shown are the percentages of (c) central memory T cells (CD44⁺CD62L⁺) and (d) effector memory T cells (CD44⁺CD62L⁻) in lymph nodes. (e) Percentages of PD-1⁺ cells among CD8⁺ T cells in lymph nodes. All the numerical data are presented as mean ± SD (0.01 < *P ≤ 0.05; **P ≤ 0.01; ***P ≤ 0.001; ****P ≤ 0.0001).

2) An important control is lacking from this study. The authors showed that their metabolic approach is much more efficient compared to conventional genetic approaches to express tags on the exosomes. But, they should then also show functionally, in vitro or in vivo, the advantages of their method, compared to conventional approaches of tagging.

Response: We appreciate this comment from the Reviewer. In the revised manuscript, we have included significant new experimental data to focus on describing our new EV metabolic labeling method that could benefit researchers in the field of EV, instead of emphasizing the comparison with conventional EV tagging methods (also due to the lack of reliable methods to quantify tags introduced via conventional methods).

For the cancer vaccination application, the key hypothesis we tested is whether the CpG-conjugated EVs show enhanced CTL response and antitumor efficacy, in comparison with EV vaccines being evaluated in clinical trials (EV alone or the mixture of EV and adjuvant). As we demonstrated in our manuscript, CpG-conjugated EVs show superior CTL response and therapeutic antitumor efficacy compared to the mixture of CpG and EVs or EV alone.

The control study suggested by the Reviewer on comparing the functionality aspect of our methods

with other chemical modification methods is beyond the scope of this work and will face inherent problems: (1) There is no gold standard regarding the functionalization of EVs, and it is nearly impossible to precisely control the number of CpG that can be conjugated across different methods. (2) There will be significant variations across laboratories regarding how one chemically modifies EVs using the protein expression method or amine-carboxyl chemistry. Any comparison data we obtain would be debatable in the field.

3) The OVA-expressing tumor models are highly immunogenic models as they express an exogenous protein. The authors showed nice results using the B16-F10 melanoma model. It would be interesting to also look if in this less immunogenic model if the presence of specific T cells such as CD8 T cells specific for the trp2 or gp100 antigens is also increase by Exo-CpG.

Response: We thank the Reviewer for this comment. We have now supplemented an experiment to analyze trp2/gp100-specific CD8⁺ T cell response after vaccination with B16F10-derived EVs, and have summarized the data as Fig. S14. By restimulating the isolated PBMCs with trp2 and gp100 peptides *ex vivo*, we were able to detect a higher number of IFN- γ ⁺ CD8⁺ T cells in mice treated with EV-CpG than in the control mice.

We have now added the data as Fig. S14, and added the following text in the revised manuscript: “We first studied whether subcutaneous injection of CpG-conjugated B16F10-derived EVs could induce the generation of B16F10-specific CD8⁺ T cells in C57BL/6 mice. At 15 or 18 days post vaccination, PBMCs were isolated from mice and restimulated *ex vivo* with trp2 and gp100 peptides. Compared to the untreated group, CpG-conjugated EVs resulted in a significantly higher number of IFN- γ ⁺ CD8⁺ T cells in PBMCs (**Fig. S14a-e**), demonstrating the ability of CpG-conjugated B16F10-derived EVs to elicit B16F10-specific CD8⁺ T cell response. The combination of CpG-conjugated EVs and anti-PD-1 was able to slightly improve the B16F10-specific CD8⁺ T cell response and the control of B16F10 tumor growth in the following prophylactic tumor study (**Fig. S14f**).”

Figure S14. CpG-conjugated B16F10-derived EVs can induce trp2/gp100-specific CD8⁺ T cell response. (a) Timeframe of the vaccination study. C57BL/6 mice (n=5 per group) were subcutaneously injected with EV-CpG or i.p. injected with anti-PD-1 on day 1. PMBCs were isolated on days 15 and 18, followed by *ex vivo* re-stimulation with the mixture of trp2/gp100 peptides. B16F10 tumor cells were inoculated on day 20. (b) Representative FACS plots of IFN- γ ⁺ CD8⁺ T cells in PBMCs on day 15. (c) Percentage of IFN- γ ⁺ cells among CD3⁺CD8⁺ T cells in PBMCs on day 15. (d) Representative FACS plots of IFN- γ ⁺ CD8⁺ T cells in PBMCs on day 18. (e) Percentage of IFN- γ ⁺ cells among CD3⁺CD8⁺ T cells in PBMCs on day 18. (f) Average B16F10 tumor volume of each group over time. All the numerical data are presented as mean \pm SD except for (f) where data are presented as mean \pm SEM (0.01 < * P \leq 0.05; ** P \leq 0.01; *** P \leq 0.001; **** P \leq 0.0001).

4) The authors demonstrated the efficacy of their Exo-CpG compared to aPD-1 alone or aPD1 + naked exosome, but they didn't try to combine their Exo-CpG to the aPD-1. It would be interesting to demonstrate the synergy of the Exo-CpG with the aPD1 treatment, for potential future combinations with immunotherapies.

Response: We thank the Reviewer for this comment. We have now studied whether the combination of EV-CpG and anti-PD-1 can further improve the trp2/gp100-specific CD8⁺ T cell response and antitumor efficacy against B16F10 melanoma. As shown in Fig. S14, compared to EV-CpG, the combination of EV-CpG and anti-PD-1 can slightly enhance the number of trp2/gp100-specific CD8⁺ T cells in PBMCs at 15 or 18 days post vaccination. In the following prophylactic tumor study, the combination of EV-CpG and anti-PD-1 can also slightly improve the tumor control in comparison with EV-CpG. Further optimization of the combination therapy, including the dose and dosing frequency of EV-CpG and anti-PD-1, is underway.

Figure S14. CpG-conjugated B16F10-derived EVs can induce trp2/gp100-specific CD8⁺ T cell response. (a) Timeframe of the vaccination study. C57BL/6 mice (n=5 per group) were subcutaneously injected with EV-CpG or i.p. injected with anti-PD-1 on day 1. PBMCs were isolated on days 15 and 18, followed by *ex vivo* re-stimulation with the mixture of trp2/gp100 peptides. B16F10 tumor cells were inoculated on day 20. (b) Representative FACS plots of IFN- γ ⁺ CD8⁺ T cells in PBMCs on day 15. (c) Percentage of IFN- γ ⁺ cells among CD3⁺CD8⁺ T cells in PBMCs on day 15. (d) Representative FACS plots of IFN- γ ⁺ CD8⁺ T cells in PBMCs on day 18. (e) Percentage of IFN- γ ⁺ cells among CD3⁺CD8⁺ T cells in PBMCs on day 18. (f) Tumor Volume (mm³) vs Time (Day) for four groups: Untreated, α -PD-1, EV-CpG, and EV-CpG + α -PD-1.

(c) Percentage of IFN- γ ⁺ cells among CD3⁺CD8⁺ T cells in PBMCs on day 18. (f) Average B16F10 tumor volume of each group over time. All the numerical data are presented as mean \pm SD except for (f) where data are presented as mean \pm SEM ($0.01 < *P \leq 0.05$; $**P \leq 0.01$; $***P \leq 0.001$; $****P \leq 0.0001$).

Minor comments:

1) Can the authors also show the control confocal images of the cancer cell lines treated without Ac4ManAZ (Fig2a-d).

Response: We thank the Reviewer for this comment. We have now added the control images of cells without Ac4ManAZ treatment in Fig 2a-d.

Figure 2. (a-d) CLSM images of (a) 4T1 breast cancer cells, (b) LS174T colon cancer cells, (c) GL261 glioblastoma cells, and (d) BxPC-3 pancreatic cancer cells, respectively, after treated with Ac4ManAZ or PBS for three days and incubated with DBCO-Cy5 (red) for 30 min.

2) How is calculated the number of Tags per exosome? (Fig 3J). Also, the number of tags is similar, but the fluorescence intensity of Cy5 on the MSC or on exosomes derived from MSC is higher than on the dendritic cells or T cells (Fig3). Is there a differential level of expression or intensity of the signal depending on the cell type from which the exosomes are derived?

Response: For our metabolic labeling method, the number of tags per exosome is estimated by calculating the number of DBCO-Cy5 that can be conjugated to the azido-tagged EVs. We first calculated the amount of Cy5 molecules on the surface of exosomes per the standard curve of Cy5 FI versus concentration, and then estimated the number of azido tags assuming 100% reaction between DBCO and azide. Since we knew the total number of exosomes added, we could estimate the number of azido groups per exosome. We have now clarified more on the calculations in the updated Figure S3.

Fig. S3. Quantification of surface azido groups per exosome. Exosomes were collected from Ac₄ManAz- or PBS-treated E.G7-OVA cells and incubated with DBCO-Cy5 for 30 min. A standard curve of Cy5 fluorescence intensity was used to calculate the amount of conjugated Cy5 molecules, as a means to estimate the number of azido groups per exosome.

3) The figure S4a is not mentioned in the text.

Response: We thank the Reviewer for this comment. We have now cited Fig. S6a (original Fig. S4a; the gating strategy for analyzing the Cy5⁺ immune cells) in the revised manuscript.

4) The methods section can be a little more detailed. For example:

- please indicate the reference of the protocol you are mentioning for the BMDC differentiation.
- how is calculated the proliferation index of the CFSE T cell in the in vitro proliferation assay?

Response: We have now added the reference (Ref. 36, 53) for BMDC differentiation. For the proliferation of CFSE-labeled T cells, when T cells divide, the CFSE fluorescence intensity will decrease. A flowjo software was then used to analyze the CFSE peaks and calculate the proliferation index. We have now added the reference (Ref. 54, 55) for analyzing the proliferation of CFSE-stained T cells.

53 Sauter, M. *et al.* Protocol to isolate and analyze mouse bone marrow derived dendritic cells (BMDC). *STAR Protoc* **3**, 101664 (2022).

54 Terrén, I., Orrantia, A., Vitallé, J., Zenarruzabeitia, O. & Borrego, F. CFSE dilution to study human T and NK cell proliferation in vitro. *Methods Enzymol* **631**, 239-255 (2020)

55 Quah, B. J. & Parish, C. R. The use of carboxyfluorescein diacetate succinimidyl ester (CFSE) to monitor lymphocyte proliferation. *J Vis Exp*, doi:10.3791/2259 (2010).

- The sequence of the OVA peptide is SIINFEKL and not SINFEKL (typo in materials and instrumentation section: “SINFEKL-MHCI tetramer was requested from the NIH Tetramer”).

Response: We have now corrected the typo.

REVIEWERS' COMMENTS

Reviewer #1 (Remarks to the Author):

For clarity, I ask authors to report transfer methodology, probing, and imaging/analysis details, including sample denaturing and reducing conditions, transfer methodology, buffers, and the sources/information of both primary and secondary antibodies for clarity. Otherwise, the authors sufficiently addressed the listed concerns, and the manuscript is acceptable for publication.

Reviewer #2 (Remarks to the Author):

I have reviewed the revised manuscript and find that authors have sufficiently addressed my referee comments.

Reviewer #3 (Remarks to the Author):

I thank the authors for addressing all the comments.

Point-by-point reply to reviewers' comments

Reviewer 1

For clarity, I ask authors to report transfer methodology, probing, and imaging/analysis details, including sample denaturing and reducing conditions, transfer methodology, buffers, and the sources/information of both primary and secondary antibodies for clarity. Otherwise, the authors sufficiently addressed the listed concerns, and the manuscript is acceptable for publication.

Response: We have added more experimental details in the revised manuscript.